# On the Role of MLP Layers in Transformer ICL with Categorical Outcomes

**Soumya Banerjee**                                    *soumya.banerjee@duke.edu*
*Department of Electrical and Computer Engineering*
*Duke University*

**Aaron Wang**                                    *aaron.wang@duke.edu*
*Department of Electrical and Computer Engineering*
*Duke University*

**William Convertino**                            *william.convertino@duke.edu*
*Department of Electrical and Computer Engineering*
*Duke University*

**Ozanan Meireles**                                    *ozanan.meireles@duke.edu*
*Department of Surgery*
*Duke University*

**Guy Rosman**                                    *guy.rosman@duke.edu*
*Department of Surgery*
*Duke University*

**Ricardo Henao**                                    *ricardo.henao@duke.edu*
*Department of Electrical and Computer Engineering*
*Duke University*

**Xiang Cheng**                                    *xiang.cheng@duke.edu*
*Department of Electrical and Computer Engineering*
*Duke University*

**Lawrence Carin**                                    *lcarin@duke.edu*
*Department of Electrical and Computer Engineering*
*Duke University*

**Reviewed on OpenReview:** *https://openreview.net/forum?id=8P4V7V1cs4*

## Abstract

We study in-context learning (ICL) with Transformers for categorical outputs $y_i$, a setting largely unexplored compared to research on real-valued $y_i$. While attention-only Transformers can, in principle, perform functional gradient descent (GD) inference for real-valued outputs, we show that categorical $y_i$ introduce a nonlinear interlayer computation. The multi-layer perceptron (MLP) layers interleaved with attention in the standard Transformer are a natural architectural component to approximate this computation, providing a concrete role for MLPs that is absent in the real-valued setting. We characterize conditions under which attention-only models can nevertheless succeed: at early layers, when all positions share similar representations, and the softmax operates in its approximately linear regime. Our theory predicts that attention-only models should degrade at greater depth and under distribution mismatch between training and testing data – predictions we confirm empirically across synthetic data, real-world image classification with domain shift, and surgical action triplet recognition. Guided by the analysis, we propose a sparse Transformer

parameterization linked to functional GD that reduces trainable parameters by roughly $50\times$ relative to an unconstrained Transformer, with minimal performance degradation. This data efficiency proves to be particularly valuable in data-limited applications, which we demonstrate through the ICL analysis of human surgical procedures.

## 1 Introduction

The standard Transformer (Vaswani et al., 2017) alternates between two types of layers: attention layers, which mix information across input positions, and multi-layer perceptron (MLP) layers, which transform the representation at each position independently. Both are ubiquitous in practice, yet recent theoretical work on in-context learning (ICL) has found that attention alone is sufficient in many settings (von Oswald et al., 2023; Cheng et al., 2024; Ahn et al., 2023) – raising the question of what MLP layers contribute.

In ICL, a model is given a "context" of $N$ labeled examples $\{(x_1, y_1), \ldots, (x_N, y_N)\}$ and asked to predict the output $y_{N+1}$ for a new ("query") input $x_{N+1}$, without any parameter updates. The Transformer implicitly infers the input-output relationship from the context and applies it to the query. Recent work has shown that when the outputs $y_i$ are real-valued, a Transformer can perform this inference entirely through attention, implementing a form of gradient descent in its forward pass (von Oswald et al., 2023; Ahn et al., 2023; Zhang et al., 2023; Cheng et al., 2024). In that setting, the MLP layers are unnecessary.

However, most practical applications of Transformers – classification, language modeling, structured prediction – involve *categorical* outputs, where $y_i$ takes one of $C$ discrete values and the model's prediction is formed through a softmax over categories. We show that this change has significant consequences for the roles of the Transformer's components. For categorical outputs, the prediction error at each layer involves an isolated softmax nonlinearity. As a result, updating the error signal between layers requires an *isolated* nonlinear transformation that attention cannot perform. The MLP layers (which can implement such nonlinearities), interleaved with attention in the standard architecture, are the natural component to carry out this computation.

Despite this theoretical expectation, we observe empirically that attention-only Transformers can still perform well on categorical ICL in some settings. In this work, we explain both phenomena: why MLP layers are in general important for categorical ICL, and why, under specific conditions, attention-only models can nevertheless succeed. The key insight is that at early layers of the Transformer, the softmax operates in a regime where it is well approximated by a linear function, and this linear approximation can be absorbed into the attention mechanism. This approximation holds when all positions share similar internal representations – a condition met at initialization and maintained when training and testing data are drawn from similar distributions. At deeper layers, or under distribution mismatch, the approximation breaks down, and MLP layers become essential.

These insights also yield a practical benefit. The analysis identifies which Transformer parameters participate in each computation, leading to a sparse parameterization – which we call the "GD" model because of its connection to functional gradient descent (GD) – that often substantially reduces the number of trainable parameters, particularly important for many real-world settings for which training data are limited.

### 1.1 Summary of Contributions

1. **A concrete role for MLP layers in categorical ICL.** We show that categorical outcomes introduce a localized nonlinear interlayer computation that attention alone cannot perform, and that the MLP is the natural architectural component available to approximate it. This role is absent for real-valued outputs, explaining why prior work found MLPs unnecessary.

2. **Conditions under which attention-only models suffice.** We derive a linear approximation to the nonlinear prediction error that is accurate for the first two layers of inference, and show it can be implemented within attention alone. We characterize when this approximation breaks down – at deeper layers and under distribution mismatch.

3. **A sparse parameterization with dramatic data efficiency gains.** Guided by the analysis, we develop a sparse Transformer parameterization that trains with substantially less data than an uncon-

strained Transformer, validated on synthetic data, real-world image classification with domain shift, and a surgical procedure dataset.

4. **New application: multi-question categorical ICL for surgery.** We extend categorical ICL to simultaneously answer multiple questions about images, performed in context. We demonstrate this on surgical action triplet recognition from the CholecT45 dataset (Nwoye et al., 2023), where labeled data are scarce and categorical outcomes naturally occur.

## 1.2 Related Work

**Mechanistic analyses of in-context learning.** A growing body of work has connected Transformer ICL to functional gradient descent (GD) in the forward pass. For linear regression, von Oswald et al. (2023); Ahn et al. (2023); Zhang et al. (2023) showed that attention-only Transformers can implement GD updates exactly, without the need for MLP layers. Cheng et al. (2024) extended this to nonlinear functions in a reproducing kernel Hilbert space, maintaining the attention-only sufficiency result. A common thread is that these studies consider real-valued outputs $y_i$, for which the prediction error is linear in the internal representation – precisely the property that makes MLPs unnecessary. Our work shows that this property breaks down for categorical outputs, where the prediction error involves a softmax nonlinearity.

**Categorical in-context learning.** The most closely related prior work is Wang et al. (2025), who also studied categorical ICL and identified the need for a nonlinear computation between attention layers. Their solution was to introduce a novel cross-attention mechanism to compute the required nonlinear function exactly. We take a different approach: rather than modifying the architecture, we analyze how the *standard* Transformer – with its existing MLP layers – can approximate this computation. We show that the MLP layers naturally serve this role, explain when a linear approximation (and hence attention alone) suffices, and leverage these insights for a sparse parameterization that dramatically reduces training data requirements. Our empirical results align well with those of Wang et al. (2025) on their synthetic benchmark, and we extend the investigation to real-world settings with distribution mismatch, where the distinction between MLP-based and attention-only models becomes pronounced.

**Sparsity and initialization.** Our sparse parameterization connects to the Lottery Ticket Hypothesis (Frankle & Carbin, 2019; Brix et al., 2020), which posits that dense networks contain sparse subnetworks that, with appropriate initialization, can match full-model performance. While lottery ticket methods discover sparsity patterns through empirical pruning, our sparsity structure is derived from analysis of what computations the Transformer needs to perform for categorical ICL. This also relates to work on solution multiplicity in neural networks (Draxler et al., 2018; Garipov et al., 2018; Lee et al., 2019), which has shown that different parameter configurations can yield similar functional behavior – helping explain why our sparse parameterization and unconstrained Transformers converge to similar predictions despite very different parameter counts.

**Applications of categorical ICL.** Most mechanistic studies of Transformer ICL have been limited to synthetic data. We demonstrate categorical ICL on real-world image classification with domain shift (training on ImageNet1k (Deng et al., 2009), testing on TinyImageNet (Le & Yang, 2015), Caltech256 (Griffin et al., 2007) and DomainNet (Peng et al., 2019)), and on surgical action triplet recognition from CholecT45 (Nwoye et al., 2023; Nwoye & Padoy, 2022). The surgical application is among the first real-world use cases of Transformer-based ICL with categorical outcomes, in a domain where labeled data are scarce and the ability to answer multiple categorical questions of an image, in context, is practically important.

## 2 Categorical ICL: From Inference to Architecture

Before describing the Transformer implementation in detail, we develop the reasoning that shows that the Transformer is a natural model for ICL, here extended to categorical $y_i$. We start from the problem of inferring a latent representation from categorical observations, and show how the key components of the Transformer – attention layers, MLP layers, and their interleaved structure – emerge naturally from the requirements of this problem.

### 2.1 The prediction error for categorical outputs

In-context learning (ICL) is characterized by a set of input-output pairs $\{(x_i, y_i)\}_{i=1}^N$, where $x_i \in \mathbb{R}^d$ are covariates, and $y_i \in \{1, \ldots, C\}$ is the categorical observation. We are also given a *query* $x_{N+1}$ for which we wish to *predict* the outcome $y_{N+1}$. Inference in this setup is characterized by inferring a latent vector $f_i \in \mathbb{R}^{d'}$ for each $i = 1, \ldots, N$, and the probability of the outcome is assumed modeled via the softmax as

$$p(Y_i = c \mid f_i) = \mathrm{softmax}(W_e^\top f_i)_c = \frac{\exp(w_c^\top f_i)}{\sum_{c'=1}^{C} \exp(w_{c'}^\top f_i)}, \tag{1}$$

where each category $c \in \{1, \ldots, C\}$ has an associated embedding vector $w_c \in \mathbb{R}^{d'}$, these vectors constitute the embedding matrix $W_e \in \mathbb{R}^{d' \times C}$ and $\mathrm{softmax}(W_e^\top f_i)_c$ is element $c$ of the $C$-dimensional vector $\mathrm{softmax}(W_e^\top f_i)$.

We wish to provide insights into how ICL inference connects to Transformers by first utilizing intuition without imposing an explicit assumption on the form of $f(x)$ that produces $f_i$ (for example, we *do not* assume it is in a reproducing kernel Hilbert space (RKHS), as in prior work Cheng et al. (2024); Wang et al. (2025)). To (implicitly) refine $f_i$, we need an error signal: a measure of the discrepancy between the model's current prediction and the observed outcome. The model predicts the probability vector $\mathrm{softmax}(W_e^\top f_i) \in (0, 1)^C$, while the observation is the one-hot vector $\mathrm{one\text{-}hot}(y_i) \in \{0, 1\}^C$. A natural prediction error is their difference:

$$\mathrm{one\text{-}hot}(y_i) - \mathrm{softmax}(W_e^\top f_i) \in (-1, 1)^C. \tag{2}$$

When the number of categories $C$ is large – as in language modeling, where $C$ can be tens of thousands – working directly with $C$-dimensional error vectors is impractical. A natural step is to project into the lower-dimensional embedding space by left-multiplying (2) by $W_e$:

$$W_e \cdot \mathrm{one\text{-}hot}(y_i) - W_e \cdot \mathrm{softmax}(W_e^\top f_i) = w_{y_i} - \mathbb{E}(w|f_i) \in \mathbb{R}^{d'}. \tag{3}$$

The first term is the embedding vector $w_{y_i}$ of the observed category, as in the way tokens are mapped to learn embedding vectors in language models (Vaswani et al., 2017). The second is

$$\mathbb{E}(w|f_i) = \sum_{c=1}^{C} w_c \cdot \mathrm{softmax}(W_e^\top f_i)_c = W_e \cdot \mathrm{softmax}(W_e^\top f_i), \tag{4}$$

the softmax-weighted average of all embedding vectors – the model's current expected embedding given $f_i$. The error in (3) lives in $\mathbb{R}^{d'}$, where typically $d' \ll C$, and it aligns with the dimension of $f_i$.

## 2.2 Sharing statistical strength via attention

The difference $w_{y_i} - \mathbb{E}(w|f_i)$ constitutes a natural direction for the change of $f_i$ so that it aligns with the embedding vector associated with $y_i$. However, as the generative process in (1) is stochastic (low-probability $y_i$ may be drawn), independent use of each $y_i$ may be inadequate (yield noisy estimates). By performing a weighted average of $w_{y_j} - \mathbb{E}(w|f_j)$ for $j = 1, \ldots, N$, we seek a smoother representation of the latent function $f(x)$, with $f_i = f(x_i)$. Specifically, for updating $f_i$, we place more weight on $w_{y_j} - \mathbb{E}(w|f_j)$ when associated covariates $x_j$ are "closer" to $x_i$ (with multiple possible definitions of "close," as discussed below). Assume there is a weight function $\kappa(x_i, x_j)$ that measures the similarity between covariates $x_i$ and $x_j$ in this manner. The weight function is designed such that $\kappa(x_i, x_j) \to 1$ smoothly as $\|x_i - x_j\|_2 \to 0$, and $\kappa(x_i, x_j) \to 0$ as $x_i$ and $x_j$ becoming increasingly dissimilar.

The above discussion motivates refining the representation at position $j$ via a weighted average of error signals across positions:

$$f_{i,\ell+1} = f_{i,\ell} + \frac{\alpha}{N} \sum_{j=1}^{N} \left[ w_{y_j} - \mathbb{E}(w|f_{j,\ell}) \right] \kappa(x_i, x_j), \tag{5}$$

where $\alpha$ is a step size, and we have introduced the index $\ell$ to denote successive refinement steps.

## 2.3 The nonlinear expectation and the role of MLPs

The update in (5) refines the representation from $f_{i,\ell}$ to $f_{i,\ell+1}$. To apply the same type of update at the next step, we need the error signal at the updated representation: $w_{y_i} - \mathbb{E}(w|f_{i,\ell+1})$. The observed embedding $w_{y_i}$ is unchanged, but $\mathbb{E}(w|f_{i,\ell+1})$ must be recomputed – and from (4), this quantity passes through the softmax and is therefore a *nonlinear* function of $f_{i,\ell+1}$.

The weighted averaging that defines (5) is a linear operation on the error signals $w_{y_j} - \mathbb{E}(w|f_{j,\ell})$, with weights $\kappa(x_i, x_j)$. The computation needed to prepare the error signal for the next step – evaluating $\mathbb{E}(w|f_{i,\ell+1})$ – is well suited to something beyond weighted averaging.

What is needed is a component that acts on each position's representation *independently* and can perform nonlinear transformations. If we interleave the weighted-averaging steps with such a component, the overall process at each step becomes:

- **Weighted averaging (attention):** Update $f_{i,\ell} \to f_{i,\ell+1}$ by accumulating error signals across positions, weighted by covariate similarity.

- **Position-wise nonlinear transformation (MLP):** Compute or approximate $\mathbb{E}(w|f_{i,\ell+1})$ from the updated representation, preparing the error signal for the next step.

This is exactly the structure of the standard Transformer: attention layers interleaved with MLP layers that act independently on each position. The reasoning here provides a concrete account of what each component contributes in the categorical setting.

For real-valued outputs $y_i$, the analogous error signal is $y_i - f_{i,\ell}$, which is linear in $f$. No interlayer nonlinear computation is needed, and attention alone suffices (Cheng et al., 2024). The MLP's role identified here is specific to categorical outputs.

We emphasize appropriate caution: we do not claim that the MLP computes $\mathbb{E}(w|f_{i,\ell+1})$ exactly. More broadly, our analysis identifies a specific nonlinear computation that the aforementioned functional update requires, and that the MLP is well suited for. We do not rule out the possibility that attention models with or without MLP could achieve good categorical ICL through mechanisms outside the functional-update framework elucidated above — but our experiments show that in practice, attention-only models degrade precisely in the conditions our analysis predicts.

When presenting results in Section 5, we will compare to Wang et al. (2025). In that work $\mathbb{E}(w|f_{i,\ell})$ is computed *exactly* with a cross-attention model. While that model provides a good point of comparison, the cross-attention form in Wang et al. (2025) does not align with the structure of the widely used Transformer.

### 2.4 Multiple similarity functions and multi-head attention

The update in (5) uses a single similarity function $\kappa(x_i, x_j)$ to weight the error signals. But there is no reason to believe that a single measure of covariate similarity captures all the relevant structure. Different aspects of $x_i$ may be informative in different ways – for instance, some dimensions of $x$ may be relevant for distinguishing certain categories, while other dimensions matter for others. This motivates using $H$ similarity functions $\kappa_h(x_i, x_j)$ operating in parallel, each with its own weighting of the error signals:

$$f_{i,\ell+1} = f_{i,\ell} + \sum_{h=1}^{H} \Lambda_{h,\ell} \sum_{j=1}^{N} \left[ w_{y_j} - \mathbb{E}(w|f_{j,\ell}) \right] \kappa_h(x_i, x_j), \tag{6}$$

where $\Lambda_{h,\ell} \in \mathbb{R}^{d' \times d'}$ allows each similarity function to contribute differently at each refinement step. This is exactly multi-head attention (Vaswani et al., 2017): each "head" $h$ computes its own similarity scores and its own weighted combination of error signals, and the results are combined.

If one further assumes that $f(x)$ resides in a reproducing kernel Hilbert space (RKHS) (Schölkopf & Smola, 2002) expressible as a superposition of $H$ kernels, then (6) can be derived formally as functional gradient descent, as stated in the following proposition.

**Proposition 1** *If the latent function $f_i = f(x_i)$ may be expressed as the superposition of $H$ kernels $\kappa_h(x_i, x_j)$, for $h = 1, \ldots, H$, then functional gradient descent (GD) for $f_i$ associated with $(x_i, y_i)$, when $y_i$ is categorical and assumed generated as in (1), yields the update in (6), where $\Lambda_{h,\ell}$ depends on kernel type $h$ and GD step $\ell$.* The proof is given in Appendix E.

The similarity functions $\kappa_h$ need not be formal kernels. A natural choice is softmax-based attention:

$$f_{i,\ell+1} = f_{i,\ell} + \sum_{h=1}^{H} \Lambda_{h,\ell} \sum_{j=1}^{N} \left[ w_{y_j} - \mathbb{E}(w|f_{j,\ell}) \right] \frac{\exp(x_i^\top x_j / \lambda_h)}{\sum_{j'=1}^{N} \exp(x_i^\top x_{j'} / \lambda_h)}, \tag{7}$$

with scalars $\lambda_h > 0$, which may be further generalized by replacing the scaled inner products $x_i^\top x_j / \lambda_h$ with $(K_h x_i)^\top (Q_h x_j)$, where $K_h \in \mathbb{R}^{d \times d}$ and $Q_h \in \mathbb{R}^{d \times d}$ are learned projection matrices.

## 3 Transformer Realization

We now show how the inference procedure developed in Section 2 maps to the standard Transformer architecture (Vaswani et al., 2017). The refinement steps become Transformer layers, the weighted averaging becomes multi-head attention, and the position-wise nonlinear transformation becomes the MLP.

### 3.1 Input encoding

For a context of $N$ labeled examples $(x_i, y_i)$ with covariates $x_i \in \mathbb{R}^d$ and categorical outcomes $y_i \in \{1, \dots, C\}$, plus a query $x_{N+1}$, the Transformer input is

$$Z_0 = \begin{bmatrix} x_1 & x_2 & \dots & x_N & x_{N+1} \\ w_{y_1} & w_{y_2} & \dots & w_{y_N} & 0_{d'} \\ 0_s & 0_s & \dots & 0_s & 0_s \end{bmatrix} \in \mathbb{R}^{(d+d'+s) \times (N+1)}. \tag{8}$$

where $0_s$ is an $s$-dimensional all-zeros vector. Each column encodes one position: the covariates $x_i$, the embedding $w_{y_i}$ of the observed category (or zeros for the query, since $y_{N+1}$ is what we seek to predict), and $s$ dimensions of scratch space that the Transformer may use for intermediate computations (Akyurek et al., 2023). Following the discussion in Section 2.1, the encoding of categorical outcomes by their embedding vectors is the natural counterpart of token embeddings in language models (Vaswani et al., 2017).

The scratch space stores the evolving representation $f_{i,\ell}$ and, when MLP layers are present, the current approximation to $\mathbb{E}(w|f_{i,\ell})$. Specifically, the vector at position $i$ and layer $\ell$ has the structure

$$z_{i,\ell}^\top = \Big[\, x_i \,,\, w_{y_i} \,,\, \underbrace{f_{i,\ell} \,,\, \mathbb{E}(w|f_{i,\ell})}_{\text{scratch space}} \,\Big], \tag{9}$$

where we initialize $f_{i,0} = 0_{d'}$ and hence $\mathbb{E}(w|f_{i,0}) = \frac{1}{C}\sum_{c=1}^{C} w_c$ for all positions $i$. *Importantly, in (9) we write $\mathbb{E}(w|f_{i,\ell})$ to connect to the above motivating construction; in practice this term is modeled via a MLP with input $f_{i,\ell}$, and it need not exactly implement the motivating expectation.*

### 3.2 Attention layers

The attention mechanism for head $h \in \{1, \dots, H\}$ is

$$\text{Attn}_h(Z) = V_h Z M \cdot A(K_h Z, Q_h Z), \tag{10}$$

where $V_h$, $K_h$, and $Q_h$ are learnable $(d+d'+s) \times (d+d'+s)$ matrices, and $A(\cdot)$ computes an $(N+1) \times (N+1)$ matrix of attention weights. Following Cheng et al. (2024), the mask $M = \begin{bmatrix} I_{N \times N} & 0_{N \times 1} \\ 0_{1 \times N} & 0 \end{bmatrix}$ ensures that only the $N$ labeled positions serve as keys and values. Column $N+1$ of $A(K_h Z, Q_h Z)$ corresponds to the query at position $N+1$ in the sequence, and attention yields an update of $f_{N+1,\ell}$, in terms of the associated data at positions $i = 1, \dots, N$. We consider both kernel-based attention (Cheng et al., 2024) and softmax-based attention (Vaswani et al., 2017); in all experiments, these yield similar results.

This realizes the weighted averaging from (5): the query $Q_h$ and key $K_h$ matrices extract the covariates $x_i$ to compute similarity scores $\kappa(x_i, x_j)$, and the value matrix $V_h$ extracts the embedding-space error $w_{y_i} - \mathbb{E}(w|f_{i,\ell})$. Multiple heads allow different aspects of this computation to proceed in parallel. The output of all heads is $\sum_{h=1}^{H} P_{h,\ell} \cdot \text{Attn}_{h,\ell}(Z_\ell)$, where $P_{h,\ell}$ (and $V_{h,l}$, $K_{h,l}$, $Q_{h,l}$ in (10)) are learnable projection matrices.

### 3.3 Attention blocks: interleaving attention and MLPs

Each Transformer layer consists of an attention block that combines multi-head attention with an MLP and skip connections:

$$Z_{\ell+1} = \tilde{Z}_{\ell+1} + \text{MLP}_\ell(\tilde{Z}_{\ell+1}) \,, \qquad \tilde{Z}_{\ell+1} = Z_\ell + \sum_{h=1}^{H} P_{h,\ell} \cdot \text{Attn}_{h,\ell}(Z_\ell), \tag{11}$$

where $\mathrm{MLP}_\ell(\cdot)$ acts independently on each column of $\tilde{Z}_{\ell+1}$. This realizes the two-step process from Section 2.3: the attention component updates $f_{i,\ell} \to f_{i,\ell+1}$, and the MLP computes or approximates $\mathbb{E}(w|f_{i,\ell+1})$ for use at the next layer. To be consistent with the above motivation, when implementing the GD-like form of the model, $f_{i,\ell}$ is used as the input to the MLP. When MLP layers are absent, $Z_{\ell+1} = \tilde{Z}_{\ell+1}$, and the model is *attention-only*; this corresponds to the linear approximation regime discussed in Section 4.

For an $L$-layer Transformer, the updates in (11) are applied sequentially for $\ell = 0, \dots, L$. The approximate representation $\hat{f}(x_{N+1})$ of $f(x_{N+1})$ is read from the scratch-space components of column $N+1$ in $Z_{L+1}$.

### 3.4 Output prediction and training

The final prediction is formed via softmax over category embeddings, as motivated in (1):

$$p(Y_{N+1} = c \,|\, x_{N+1}, \mathcal{C}) = \frac{\exp[w_c^\top \hat{f}(x_{N+1})]}{\sum_{c'=1}^C \exp[w_{c'}^\top \hat{f}(x_{N+1})]}. \tag{12}$$

The Transformer parameters – including the embedding vectors $\{w_c\}$ – are learned by minimizing the cross-entropy loss over a training set of $M$ contextual datasets $\{(x_i^{(m)}, y_i^{(m)})\}_{i=1}^{N+1}$ for $m = 1, \dots, M$, and $\mathcal{C} = \{(x_i, y_i)\}_{i=1}^N$ is the context dataset for $x_{N+1}$.

### 3.5 Sparse "GD" parameterization

The analysis in Section 2 identifies a specific role for each Transformer component. This motivates a sparse parameterization in which only the parameters relevant to these roles are free, and all others are set to zero. Concretely:

- The query and key matrices $Q_{h,\ell}$ and $K_{h,\ell}$ are constrained to operate only on the covariate subspace $(x_i)$, computing covariate similarity.

- The value matrix $V_{h,\ell}$ is constrained to extract the embedding-space error $w_{y_i} - \mathbb{E}(w|f_{i,\ell})$.

- The MLP at each layer receives $f_{i,\ell+1}$ and outputs (in principle) an approximation to $\mathbb{E}(w|f_{i,\ell+1})$.

The detailed parameter constructions, for both the MLP and attention-only variants, are provided in Appendices F and I, respectively. The resulting models have far fewer trainable parameters than an unconstrained Transformer – for instance, approximately 300 *vs.* 15,000 for a three-layer model with MLPs on our synthetic benchmark (see Appendix A for full parameter counts across all experiments).

Following convention in the literature (Cheng et al., 2024; von Oswald et al., 2023; Wang et al., 2025), we refer to this sparse parameterization as the "GD" model (recognizing, as discussed above, that it is not necessarily exactly gradient descent). As in the literature, when the Transformer is trained without constraints it is referred to as "Trained TF."

## 4 Linear Approximation: Attention-Only Transformer

Let $f_{i,\ell} = f_{i,\ell-1} + \Delta f_{i,\ell-1}$, and therefore $\Delta f_{i,\ell-1} = \frac{\alpha}{N} \sum_{j=1}^N (w_{y_j} - \mathbb{E}(w|f_{j,\ell-1}))\kappa(x_i, x_j)$ is the correction to the inferred vector after the layer-$\ell$ attention (with $H = 1$ for simplicity). In Appendix H we show that under a first-order Taylor expansion

$$\mathbb{E}(w|f_{i,\ell}) \approx \mathbb{E}(w|f_{i,\ell-1}) + W_e \cdot \mathrm{diag}[\mathrm{softmax}(W_e^\top f_{i,\ell-1})] \cdot \tilde{W}_{e,\ell-1}^\top \cdot \Delta f_{i,\ell-1}, \tag{13}$$

recalling that the $c$th column of $W_e \in \mathbb{R}^{d' \times C}$ is the embedding vector $w_c$; the $c$th column of $\tilde{W}_{e,\ell-1}$ is $w_c - \mathbb{E}(w|f_{i,\ell-1})$.

With the *same* initialization as discussed above, $f_{i,0} = 0_{d'}$, $\mathrm{softmax}(W_e^\top f_{i,0})$ is uniformly distributed for all $i$, and therefore we may approximate

$$\mathbb{E}(w|f_{i,1}) \approx \frac{1}{C}\Big[ \sum_{c=1}^C w_c + W_e \tilde{W}_e^\top \Delta f_{i,0} \Big] = \mathbb{E}(w|f_{i,0}) + \frac{1}{C} W_e \tilde{W}_e^\top \Delta f_{i,0}, \tag{14}$$

where $\Delta f_{i,0}$ is the output of the preceding (first) self-attention layer, and $\tilde{W}_e$ is *independent of* index $i$. As shown in Appendix I, the linear update of the expectation in (14), represented by $\frac{1}{C}W_e\tilde{W}_e^\top \Delta f_{i,0}$, can be implemented within the same self-attention layer used to compute $\Delta f_{i,0}$, and therefore there is not a need for an MLP layer for its computation (*within the linear approximation*).

There are two reasons that the approximation in (14) could work well: (*i*) $f_{i,0} = 0_{d'}$ is the same for all $i = 1, \ldots, N$, and therefore the *same* matrix $\frac{1}{C}W_e\tilde{W}_e^\top$ holds in the linear approximation for all $i$; (*ii*) the linear approximation is performed about $f_{i,0} = 0_{d'}$, which is the *center* of the linear region of the softmax, for *all* $c = 1, \ldots, C$.

**The first *two* steps of functional refinement typically can be performed well with self-attention alone, *assuming a good match between the training and testing data*.** With the initialization $f_{i,0} = 0_{d'}$, the first GD step can be done *exactly* with the first self-attention layer (note that setting $f_{i,0} = 0_{d'}$ is not special to this setting; it aligns with all prior ICL research (von Oswald et al., 2023; Cheng et al., 2024; Wang et al., 2025; Ahn et al., 2023; 2024)). Based on the above discussion, with that same first attention layer, one may accurately update $\mathbb{E}(w|f_{i,0}) \to \mathbb{E}(w|f_{i,1})$ within a linear approximation (with linear approximation that is the same for all positions $i$). The next (second) self-attention layer, leveraging the updated $\mathbb{E}(w|f_{i,1})$, can then perform the second step to compute $\Delta f_{i,1}$. Hence, it is anticipated that the first *two* steps of functional GD inference can be performed well with two self-attention layers, and *no* MLPs.

**After the first two self-attention layers, the linear approximation is less appropriate.** From (13), the linear approximation for $\mathbb{E}(w|f_{i,2})$ involves an expansion about $f_{i,1}$. There are two problems with this: (*i*) In general $f_{i,1}$ is *different* for each $i$, and therefore the matrix $W_e \cdot \text{diag}[\text{softmax}(W_e^\top f_{i,1})] \cdot \tilde{W}_{e,1}$ in (13) used in a linear approximation *depends on* $i$ (but in an attention-only Transformer, we *have* to assume the same linear relationship for all $i$); (*ii*) depending on $f_{i,1}$, the softmax function may no longer be in its linear regime, further undermining the linear approximation. Therefore, for $\mathbb{E}(w|f_{i,\ell})$ for $\ell \geq 2$, which are needed for GD steps 3 and beyond, a *nonlinear* representation of the expectation may be important.

Importantly, it is possible that this linear approximation may be sensitive to a match between the training and testing data, such that the *same* (learned) linearization holds for all test data. Under distribution mismatch, the update $\Delta f_{i,\ell}$ may be systematically larger or more variable than what the model encountered during training, as the learned similarity function $\kappa(x_i, x_j)$ may assign inappropriate weights to out-of-distribution covariates. This can push $f_{i,\ell}$ out of the softmax's linear regime, causing the linear approximation to break down even at the second layer. We examine this phenomenon in detail in subsequent experiments.

These observations do not constitute a formal impossibility result for attention-only models — an attention-only Transformer could in principle learn a different algorithm that does not require explicitly recomputing $\mathbb{E}(w|f_{i,\ell})$. However, the functional GD framework predicts specific failure modes for attention-only models (deeper layers, distribution mismatch), and if these predictions are borne out empirically, it provides strong evidence that the GD-like computation is indeed what these models learn — and that the MLP's role in supporting it is practically important. In Appendix H the linear approximation to the MLP is developed in detail, and in Appendix I we show how such can be implemented by an attention-only model.

## 5 Experiments

We examine four questions: (*i*) how data-efficient is the sparse GD parameterization relative to an unconstrained Trained TF? (*ii*) when do MLP layers matter, and does distribution mismatch predict their necessity? (*iii*) does the Trained TF learn an algorithm similar to the GD model? (*iv*) can categorical ICL be applied to real-world problems?

In all experiments, we consider both kernel-based (RBF) and softmax-based attention. These yield similar results throughout; we report results with softmax attention, and results with RBF attention are nearly identical to these[1]. Five random seeds are used for each model, and error bars reflect variance across seeds. Full parameter counts for all models are provided in Appendix A. In all experiments, $H = 2$ attention heads are used, and the category embedding-vector matrix $W_e$ is learned with all other parameters.

---

[1]Code: https://github.com/SouBanerjee/code-tf-gd-mlp

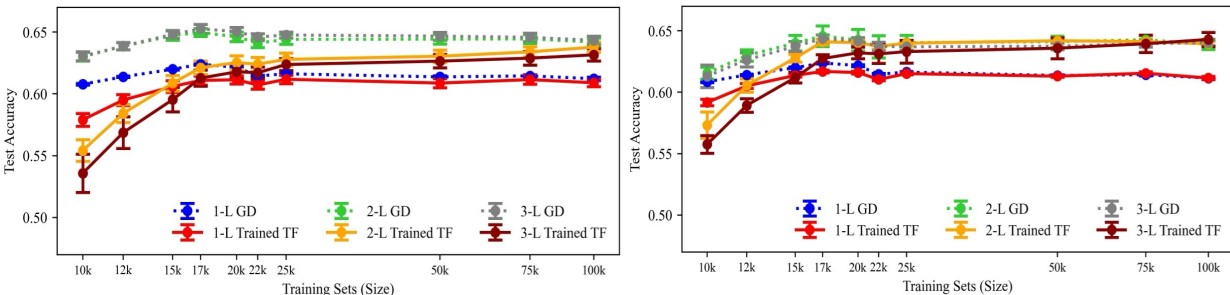

Figure 1: Classification accuracy on the synthetic data ($C = 25$ categories, $N = 50$ context samples) as a function of the number of training contexts $M$. Left: Transformers with MLP layers. Right: attention-only Transformers (no MLPs). Results are shown for the GD (sparse) and Trained TF (unconstrained) parameterizations, for 1, 2, and 3 layer models (*e.g.*, 3-L denotes three layers). Error bars reflect variance across five random seeds.

## 5.1 Synthetic data: data efficiency and the role of MLPs

We begin with the synthetic data introduced in Wang et al. (2025), which represents a challenging $C = 25$ category in-context classification problem, $x_i \in \mathbb{R}^{10}$ and $N = 50$. For convenience, details of the synthetic data generation are provided in Appendix B. Figure 1 presents classification accuracy as a function of the number of training contexts $M$, for both MLP and attention-only Transformers, under the GD and Trained TF parameterizations. We highlight four findings.

**The sparse GD model is dramatically more data-efficient than Trained TF.** Across all model configurations in Figure 1, the GD parameterization reaches its peak performance with roughly $M = 20,000$ training contexts, while Trained TF requires $M = 100,000$ or more to match this level. This is a direct consequence of the parameter reduction identified in Section 3.5: for a three-layer model with MLPs, GD has approximately 300 trainable parameters compared to roughly 15,000 for Trained TF (see Appendix A).

**Deeper models improve performance.** For both GD and Trained TF, moving from one to two to three layers yields consistent improvements, reflecting the benefit of iterative refinement as described in Section 2. The gains are most pronounced for the GD models, which can exploit the additional layers effectively even with limited training data.

**When training and testing distributions match, attention-only models perform nearly as well as MLP models.** Comparing the left and right panels of Figure 1, the attention-only models (right) achieve classification accuracy very close to their MLP-equipped counterparts (left). This is consistent with the analysis in Section 4: when training and testing data are drawn from the same distribution (as they are here), the linear approximation to $\mathbb{E}(w|f_{i,\ell})$ remains accurate, and the MLP layers provide little additional benefit.

**GD and Trained TF converge to similar performance.** When Trained TF is given sufficient data ($M \geq 100,000$), its accuracy closely matches GD. This holds for both MLP and attention-only models, suggesting that both parameterizations converge to similar inference algorithms – a hypothesis we examine directly in Section 5.4. We also note that these results align well with those reported in Wang et al. (2025) on the same synthetic benchmark (in which $\mathbb{E}(w|f_{i,\ell})$ was computed exactly with cross-attention); detailed comparisons are presented in Appendix C.

## 5.2 Distribution mismatch: when MLP layers become essential

The synthetic experiment above uses training and testing data drawn from the same distribution. We now examine what happens when this assumption is violated, by training on one image dataset and testing on others. The covariates $x_i$ are features from a pre-trained masked-autoencoder vision Transformer (He et al., 2022; Dosovitskiy et al., 2020), yielding $d = 768$ dimensional representations. We use embedding vectors of dimension $d' = 5$. Each context contains $N = 50$ samples from $C = 5$ randomly selected categories, and the query is from one of these categories. The Transformer is trained on ImageNet1k (Deng et al., 2009) (1000 object categories, 700-1300 samples each) and tested on TinyImageNet (Le & Yang, 2015) (200 classes, 500 samples per category at $64 \times 64$ resolution), Caltech256 (Griffin et al., 2007) (256 object categories, 31–80 samples each) and DomainNet (Peng et al., 2019) (345 categories across five visual domains). Downsampling to $64 \times 64$ suppresses high-frequency detail and alters the image statistics, so TinyImageNet is distributionally

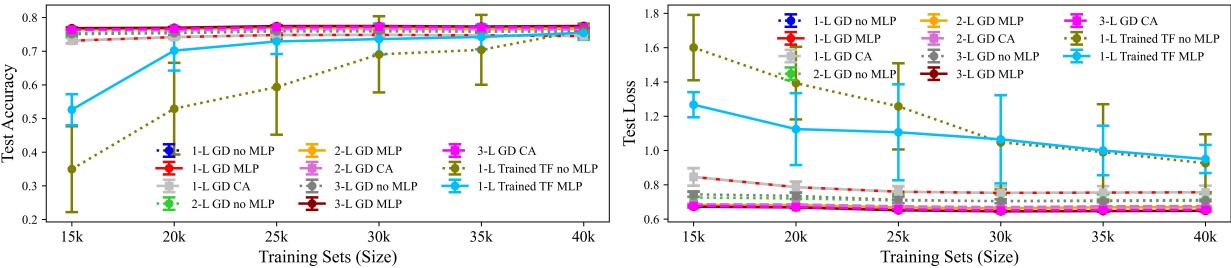

Figure 2: ICL classification accuracy (left) and loss (right) on TinyImageNet (test-data), with the Transformer models trained on ImageNet1k (training-data), as a function of the number of training contexts $M$. All GD models use sparse parameterization guided by the analysis in Section 2. Results are shown for 1, 2, and 3 layer(s) GD models. For Trained TF with or without MLP, results only shown for 1 layer model, demonstrating that it closely aligns with GD models with more data (training a 2 or 3 layer Trained TF model requires substantially more data, *i.e.*, much larger $M$). Partial zoomed-in plot comparing performance across GD models is provided in Sec J.

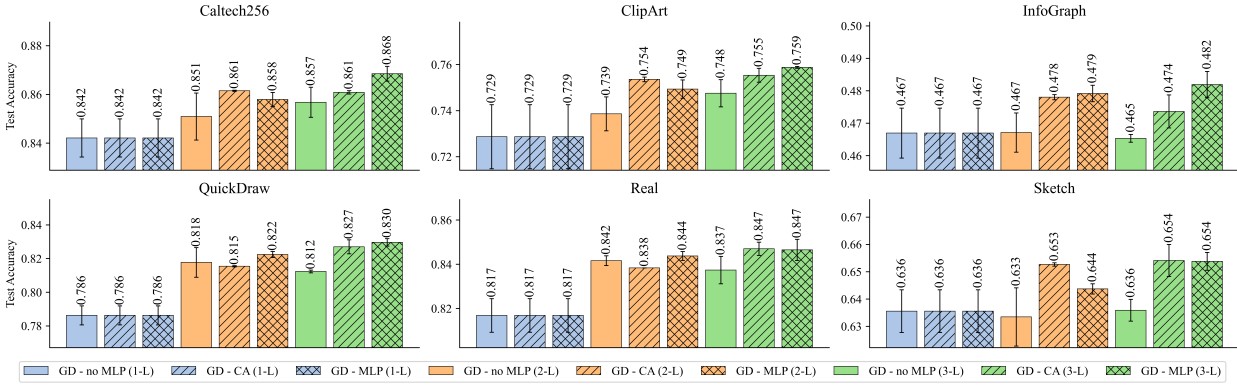

Figure 3: ICL classification accuracy on Caltech256 (test) and DomainNet (test), with the Transformer trained on ImageNet1k. Results are shown for Caltech256 and five visual domains (ClipArt, InfoGraph, QuickDraw, Real, Sketch), comparing GD with or without MLP layers and the cross-attention (CA) model of Wang et al. (2025). Error bars reflect standard deviation across multiple parameter initializations.

distinct from ImageNet1k despite its classes being a subset. Given the data efficiency advantage established above, all results here use the sparse GD parameterization. We also compare against the cross-attention (CA) model of Wang et al. (2025), which computes $\mathbb{E}(w|f_{i,\ell})$ exactly via a non-standard cross-attention mechanism (as discussed in Section 2.3). This provides a reference point for exact functional GD inference. Furthermore, we compare with Trained TF with or without MLP for 1 layer model, demonstrating that it closely aligns with GD model (1 layer) if trained with increasing number of training set.

Figures 2 and 3 reveal a striking contrast with the synthetic results. Under distribution mismatch, the attention-only model (linear approximation) degrades substantially for two- and three-layer models, while the MLP-equipped model maintains strong performance across all depths. This is precisely the pattern predicted by the analysis in Section 4: when training and testing distributions differ, the first-layer update $\Delta f_{i,0}$ may push $f_{i,1}$ out of the softmax's linear regime, causing the linear approximation to break down at the second layer and beyond.

Three specific observations are worth noting. First, the GD model with MLP layers and the CA model of Wang et al. (2025) produce highly similar results across all settings (Figures 2 left and center, and the corresponding bars in Figure 3). This indicates that the MLP in the standard Transformer effectively serves the same functional role as the exact cross-attention mechanism – consistent with our analysis that the MLP approximates $\mathbb{E}(w|f_{i,\ell})$. Second, the degradation of the attention-only model grows with depth: the one-layer model is competitive, but the two- and three-layer models fall behind significantly. This matches the prediction that the linear approximation holds for the first two refinement steps but breaks down thereafter.

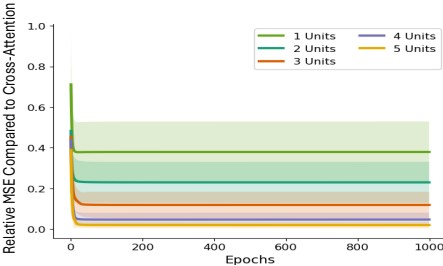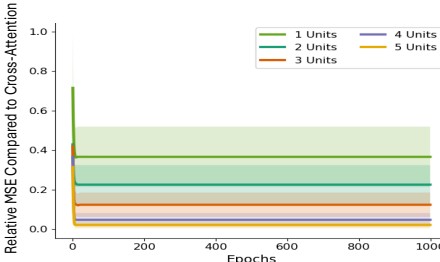

Figure 4: The cross-attention-based Transformer (Wang et al., 2025) was trained on the synthetic data studied in Figure 1 and all parameters frozen (with which $\mathbb{E}(w|f_{i,\ell})$ is computed exactly). The cross attention was removed, and replaced by an MLP (left) or with a linear approximation (right). The parameters of the MLP/matrix were then trained, with all other parameters unchanged. Here is plotted the MSE error relative to the norm of the output of the cross-attention output.

Third, the pattern is consistent across all five DomainNet domains (Figure 3), despite their very different visual characteristics (photographs, clipart, sketches, infographics, quick-draw images), confirming that the phenomenon is driven by distribution mismatch rather than properties of any particular domain.

As a reference for the difficulty of this ICL task, a $k$-nearest neighbor classifier on the same masked-autoencoder features achieves approximately 40% accuracy on TinyImageNet (best results were with $k = 7$, yielding 42% accuracy, but $k = 3$ through $k = 10$ yielded similar results).

## 5.3 Further examination of the role of MLP and the Linear Approximation

We have postulated a role for the MLP units within a Transformer, when $y_i$ is categorical, could be for computing the expectation $\mathbb{E}(w|f_{i,\ell})$. As a test for that, we consider the cross-attention-based Transformer of Wang et al. (2025), which effectively performs exact functional GD, with $\mathbb{E}(w|f_{i,\ell})$ computed exactly via cross attention. Such a Transformer was trained on the synthetic data of Section 5.1 (data generation as in Wang et al. (2025)). After training, all model parameters were frozen, and the cross-attention was removed. In place of the cross attention, we dropped in an MLP, and only trained the MLP parameters. Similarly, we dropped in a matrix approximation, and only trained the matrix. We wish to consider the degree to which the MLP/linear approximation recover the performance of the original cross-attention-based Transformer.

In Figure 4 we show results as a function of the number of hidden units in the MLP, which for the matrix approximation corresponds to the matrix rank. We observe that with five units (full rank matrix approximation), the MLP and linear approximation emulate the expectation accurately, in the relative MSE error compared to the output of the cross-attention diminishes quickly with increasing number of units in the MLP/linear models. These results are for a two-layer model. As discussed in Section 4, this is the regime for which we expect the linear approximation to work well.

## 5.4 What algorithm does Trained TF learn?

The results in Section 5.1 show that GD and Trained TF converge to similar *predictive* performance when the latter has sufficient data. However, similar predictions do not imply similar algorithms. We now examine whether the two models also learn similar internal computations.

With softmax attention, the Transformer's behavior is determined by the matrix products $Q_\ell^\top K_\ell$ (which governs attention weights) and $P_\ell V_\ell$ (which governs how values are combined), rather than by the individual matrices $Q_\ell, K_\ell, V_\ell, P_\ell$. Figure 5 visualizes these products for a two-layer attention-only Transformer trained on synthetic data, comparing GD and Trained TF (the latter trained with $M = 100,000$ contexts).

The agreement is substantial. In both models, $Q^\top K$ operates primarily on the covariate subspace ($x_i$), confirming that attention weights are determined by covariate similarity. The *PV* products show that the latent representation $f_{i,\ell}$ is updated at both layers, and the expectation is updated at the output of layer 1, consistent with the structure described in Section 2. While the agreement is not exact – the models have different parameterizations and were trained differently – the qualitative structure is strikingly similar, indicating that the Trained TF discovers an algorithm closely related to the one the GD parameterization was designed to implement.

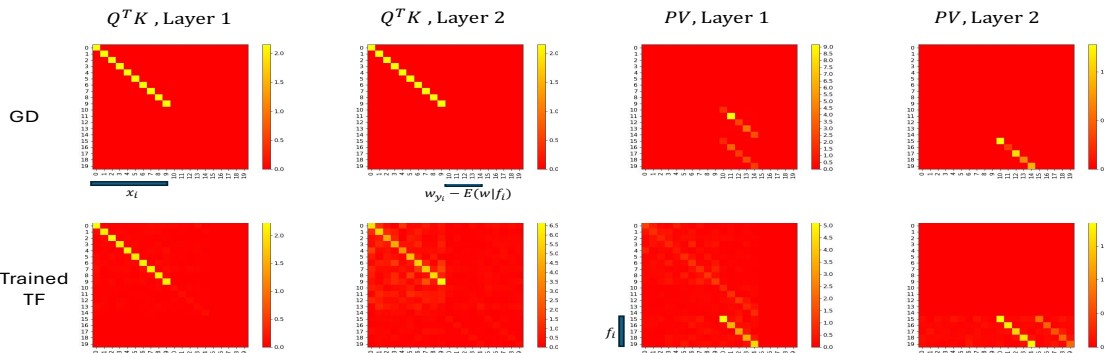

Figure 5: Learned parameter matrix products for attention-only two-layer Transformers on the synthetic data. Top row: GD. Bottom row: Trained TF. Left two columns: $Q^\top K$ at layers 1 and 2, which determines attention weights. Right two columns: $PV$ at layers 1 and 2, which determines how values are combined. The axes identify which components of the encoding $- x_i$, $w_{y_i} - \mathbb{E}(w|f_{i,\ell})$, and $f_{i,\ell}$ – are involved.

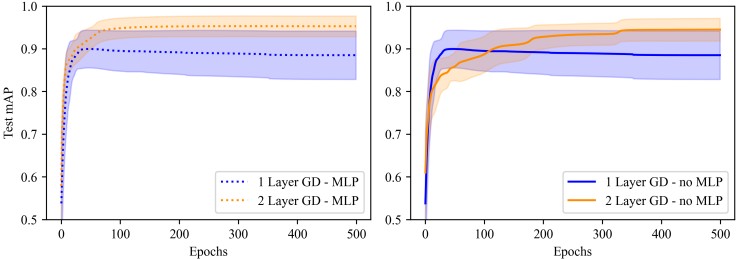

Figure 6: ICL performance on the CholecT45 surgical dataset, measured by mean average precision (mAP). Left: GD with MLP layers. Right: GD attention-only (no MLPs). Models are evaluated on action triplets not seen during training. Three random seeds are used; shaded regions indicate standard deviation.

We emphasize that this level of agreement only emerges when the Trained TF has access to a large training set ($M \geq 100{,}000$), consistent with the predictive convergence observed in Figure 1. Results are shown for the attention-only model because it can be implemented with a single attention head (see Appendix I), making the parameter comparison unambiguous. Similar agreement is observed when MLP layers are present, though the two-head structure makes the comparison more involved.

## 5.5 Real-world application: surgical action triplet recognition

We now apply categorical ICL to a real-world problem: recognizing surgical action triplets in laparoscopic video. This experiment serves two purposes. First, it demonstrates the practical utility of categorical ICL in a domain where labeled data are scarce. Second, it validates the *other* theoretical prediction from Section 4: that attention-only models should perform well when training and testing data are drawn from similar distributions.

We use the CholecT45 dataset (Nwoye et al., 2023; Nwoye & Padoy, 2022), a subset of CholecT50, consisting of 45 video recordings of laparoscopic cholecystectomy. Surgical activities are formalized as ⟨*instrument, verb, target*⟩ triplets, composed from 6 instruments, 10 verbs, and 15 targets, yielding 100 possible action triplet classes. We consider the 25 most prevalent triplets. Example images are shown in Appendix D.

Unlike the image classification experiments in Section 5.2, where the training images (Caltech256) and testing images (TinyImageNet, DomainNet) differ fundamentally in visual character, here both training and testing images come from surgeries of the same general type. The triplet *categories* seen during training differ from those at testing (we train on 10 randomly selected triplets and test on the remaining 15), but the images themselves share the same visual domain. This represents a setting with matched image distributions but novel categories.

This experiment uses the multi-question extension of categorical ICL, where $Q = 5$ binary questions (corresponding to the presence or absence of specific triplets) are answered simultaneously for each image. The update rule generalizes naturally, as described in Appendix G:

$$f_{i,\ell+1} = f_{i,\ell} + \frac{\alpha}{N} \sum_{j=1}^{N} \frac{1}{Q} \sum_{q=1}^{Q} \left[ w_{y_j^{(q)}}^{(q)} - \mathbb{E}(w^{(q)}|f_{j,\ell}) \right] \kappa(x_i, x_j), \tag{15}$$

where $w_{y_j^{(q)}}^{(q)}$ is the embedding vector for the answer to question $q$ at position $j$. The feature extractor is the masked autoencoder as in Section 5.2 ($d = 768$), with embedding dimension $d' = 4$ and context size $N = 50$.

Figure 6 shows that both the MLP and attention-only models achieve strong performance (mAP > 0.9 with two layers), with the MLP model only slightly outperforming the attention-only variant. This contrasts sharply with the image classification results in Section 5.2, where the attention-only model degraded substantially. The difference is explained by our analysis: here, the training and testing images share the same visual domain (laparoscopic surgery), so the learned similarity function $\kappa(x_i, x_j)$ generalizes well to test data, $\Delta f_{i,0}$ remains in the expected range, and the linear approximation to $\mathbb{E}(w|f_{i,\ell})$ remains accurate. In the image classification experiments, the visual domain changed entirely between training and testing, violating this condition.

## 6    Discussion and Conclusions

We have studied in-context learning with Transformers for categorical outputs. Our analysis identifies a specific, interpretable role for MLP layers in categorical ICL that is absent in the real-valued case, and our experiments demonstrate that this role becomes practically important — and the absence of MLPs practically costly — under distribution mismatch and at greater model depth

We also characterized when MLP layers are *not* needed. At early layers, when all positions share similar internal representations and the softmax operates in its approximately linear regime, the nonlinear expectation can be absorbed into the attention mechanism. This linear approximation is accurate for the first two refinement steps under matched training and testing distributions, but breaks down at deeper layers or under distribution mismatch. These predictions were validated on synthetic data, real-world image classification with domain shift, and surgical action triplet recognition.

The analysis further yielded a sparse "GD" parameterization that reduces the number of trainable parameters by roughly $50\times$ relative to an unconstrained Transformer, with minimal loss in performance. This data efficiency is particularly valuable in domains where labeled data is scarce, as demonstrated in the surgical application, and suggests that theory-guided parameter reduction may be a productive alternative to post-hoc pruning methods.

Several limitations should be noted. First, our analysis characterizes what the MLP *can* compute in the categorical ICL setting, not what a trained MLP provably learns. The empirical evidence – including the close agreement between the MLP-based and cross-attention models (Figures 2 and 3), the parameter structure learned by Trained TF (Figure 5), and the MLP replacement experiment in Appendix 5.3 – is consistent with our account, but a formal proof that trained MLPs converge to approximating $\mathbb{E}(w|f_{i,\ell})$ remains open. Second, our experiments use relatively small Transformers (1–3 layers, small embedding dimensions) in controlled ICL settings. Whether the same roles persist in large-scale language models – where the number of layers, the complexity of the data, and the training dynamics are vastly different – is an important question that our framework does not directly address, though the core observation (categorical softmax introduces a nonlinearity that attention cannot resolve) is architecture-scale independent. Third, the sparse parameterization is currently derived by hand from the analysis; automating the identification of such structure, or combining it with learned pruning, could yield further gains.

## Acknowledgments

This research was supported in part by the Office of Naval Research, under grant 313000130, and by the Duke University Department of Surgery.

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

**Table of Contents for Appendix**

## A   Parameter counts for experiments

Table 1: Comparison of total number of parameters between GD and Trained TF, with or without MLP, on the synthetic dataset.

| Layers | GD-Linear Approximation | TF-Linear Approxmation | GD-MLP | TF-MLP |
|--------|-------------------------|------------------------|--------|--------|
| 1 | 156 | 1725 | 296 | 5290 |
| 2 | 186 | 3325 | 301 | 10290 |
| 3 | 216 | 4925 | 306 | 15290 |

Table 2: Comparison of total number of parameters between GD - MLP and GD - Linear Approximation, on the image dataset considered in Figure 2.

| Layers | GD-Linear Approximation | GD-MLP |
|--------|-------------------------|--------|
| 1 | 56 | 146 |
| 2 | 86 | 151 |
| 3 | 116 | 156 |

Table 3: Comparison of total number of parameters between GD - MLP and GD - Linear Approximation, on the surgery dataset.

| Layers | GD-Linear Approximation | GD-MLP |
|--------|-------------------------|--------|
| 1 | 105 | 1245 |
| 2 | 109 | 1249 |
| 3 | 113 | 1253 |

## B   Data generation processes for synthetic data

The data are generated

$$p(Y = c | f(x)) = \exp[w_c^\top f(x)] / \sum_{c'=1}^{C} \exp[w_{c'}^\top f(x)], \tag{16}$$

Table 4: Comparison of the softmax probability over categories, at the output of Transformer ICL. At right, as a reference, we show the entropy of the softmax predictions, based on the cross-attention (CA) Transformer (Wang et al., 2025), that effectively performs exact functional GD. We also consider the *cross-entropy* between the output of Transformers considered here, relative to the softmax outputs of CA (Wang et al., 2025). The cross attention is shown between a GD-based form of the Transformer and with Trained TF (TF). Models of 1, 2 and 3 layers are considered, with the self-attention-only version of the ICL Transformer, and with MLP layers.

| Model | GD vs CA | TF vs CA | Entropy of CA |
|---|---|---|---|
| Self-Attention Only (1 Layer) | $1.377 \pm 0.062$ | $1.442 \pm 0.089$ | $1.238 \pm 0.035$ |
| Self-Attention Only (2 Layers) | $1.310 \pm 0.043$ | $1.282 \pm 0.038$ | $1.084 \pm 0.031$ |
| Self-Attention Only (3 Layers) | $1.244 \pm 0.056$ | $1.234 \pm 0.052$ | $1.028 \pm 0.038$ |
| MLP layers present (1 Layer) | $1.357 \pm 0.033$ | $1.442 \pm 0.091$ | $1.238 \pm 0.035$ |
| MLP layers present (2 Layers) | $1.268 \pm 0.049$ | $1.332 \pm 0.038$ | $1.084 \pm 0.031$ |
| MLP layers present (3 Layers) | $1.204 \pm 0.051$ | $1.271 \pm 0.051$ | $1.028 \pm 0.038$ |

for $C = 25$ and $w_c \in \mathbb{R}^5$, where $w_c$ represents the category-dependent embedding vectors used for data synthesis (hidden from the Transformer). For data synthesis, $w_c$ are generated (once) randomly, with each matrix component drawn i.i.d. from $\mathcal{N}(0, 1)$. After $W_e$ is so drawn, different contextual datasets consider a distinct function $f^{(m)}(x)$, where $m$ represents the context index. To constitute $f^{(m)}(x)$, 5 categories are selected uniformly at random from the dictionary of $C = 25$ categories. Let $c^{(m)}(1), \ldots, c^{(m)}(5)$ denote these categories for context $l$. We further randomly generate 5 respective "anchor positions," $\tilde{x}(1), \ldots, \tilde{x}(5)$, each drawn i.i.d. from $\mathcal{N}(0_d, I_d)$, where $d = 10$ (for covariates $x \in \mathbb{R}^{10}$). The function for context $m$ is represented as

$$f^{(m)}(x) = \lambda \sum_{k=1}^{5} w_{c(k)} \kappa_{RBF}[x - \tilde{x}(k); \sigma_\ell], \tag{17}$$

where the RBF kernel parameter $\sigma_m$ for component $m$ is selected such that

$$\kappa_{RBF}[x - \tilde{x}(m); \sigma_m] = \exp(-\sigma_m^2 \|x - \tilde{x}_m\|_2), \tag{18}$$

equals 0.1 at the center of the other kernel to which it is closest (in a Euclidean distance sense). Parameter $\lambda = 10$, selected so as to have category $c(m)$ be clearly most probable in the region of $\tilde{x}(m)$. Each contextual block considers $N = 50$ samples.

## C  Comparison of predictions relative to Wang et al. (2025)

In Figure 1 we considered predictions on the synthetic data introduced in Wang et al. (2025). Here we examine the similarity of predictions for the cross-attention-based ICL model of Wang et al. (2025) on these data relative to our model, with and without the MLP layers. Rather than comparing just the output predictions, which reflect top-1 predictions at the softmax output of the model, we compare the full softmax-generated probability mass function (PMF). Specifically, we treat the predictions of the cross-attention (CA) model of Wang et al. (2025) as "ground truth," because they effectively correspond to exact functional GD. We calculate the cross-entropy between the PMF generated by the CA model of Wang et al. (2025) to the output PMF from our model, with and without MLP layers. The results are summarized in Table 4.

By considering the cross-entropy between the CA-based PMF and the PMFs of our model, we examine how closely our generated PMFs align with predictions from exact functional GD. Our GD models are *guided* by functional GD analysis, but the MLP layers are sufficiently flexible to possibly perform better than functional GD inference.

Multiple issues can be examined by evaluating Table 4:

- Comparing results for GD vs CA and TF vs CA, we can evaluate the similarity of the GD and Trained TF models, respectively (comparing the GD vs CA column to the TF vs. CA column).
- For either the GD or Trained TF model, we can compare variation with and without the MLP layers (comparing rows 1-3 to rows 4-6).

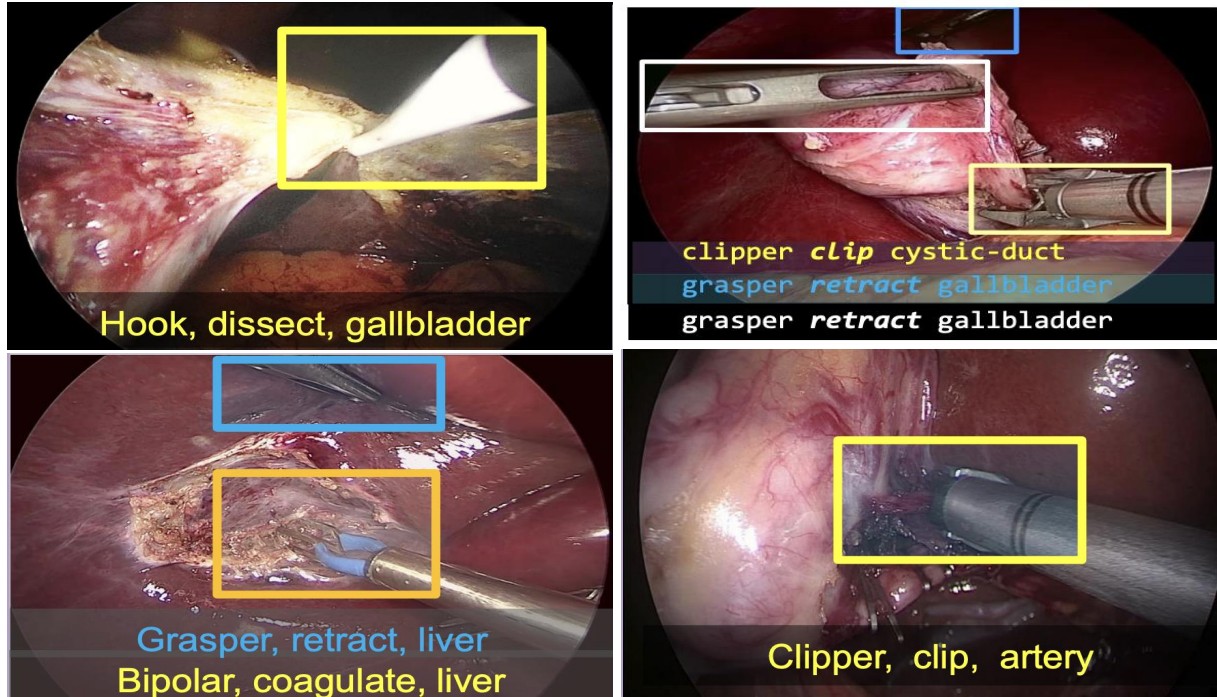

Figure 7: Example figures taken from CholecT45 dataset (Nwoye et al., 2023; Nwoye & Padoy, 2022), demonstrating surgical images with their associated action triplets. Bounding boxes are added manually for better understanding.

- The entropy of the CA model serves as a lower bound for the cross entropy.

The results in Table 4 indicate that the Trained TF softmax output is (on average) further from exact functional GD (the CA model Wang et al. (2025)) than is the GD-based design of our Transformer. However, the differences are within the standard deviation of the experiments. These results suggest that while the Trained TF model (trained without constraints) yields inference predictions, here across all 25 elements of the softmax output, that are relatively close to exact functional GD. We also note that for these simulated data, the full softmax output from the Transformer with and without MLPs are similar, with differences within the standard deviation. However, we emphasize that one cannot expect the self-attention-alone models to be sufficient in general.

## D    Example images from the surgery dataset

Example images from the surgery dataset are shown in Figure 7. The rectangles identify where in the image a "triplet" resides, and the three words associated with the triplet are also depicted. Note that the number if triplets in an image is greater than or equal to one, and the ICL algorithm is tasked with identifying the presence/absence of all triplets in a given image. Note that the triplets are diverse, and that our ICL Transformer is trained on one set of possible triplets, and tested on a distinct set. While the *type* of triplets between training and testing data are distinct, all images are connected to surgery, so there is not a mismatch on the form of the images.

## E    Proof of Proposition 1

Our derivation is based on the assumption that $f(x)$ resides in a reproducing kernel Hilbert space (RKHS) (Schölkopf & Smola, 2002), but the setup extends to softmax-based attention kernels as well (Wang et al., 2025). From the RKHS perspective, let $f(x) = A\psi(x)$, with $\psi(x)$ a *fixed* mapping of covariates $x$ to a Hilbert space, and the parameters acting in that space are $A$.

The cross-entropy cost function for inferring the parameters $A \in \mathbb{R}^{d' \times m}$ may be expressed as

$$\mathcal{L}(A) = -\frac{1}{N} \sum_{i=1}^{N} \log \left[ \frac{\exp[w_{y_i}^\top (A\psi(x_i))]}{\sum_{c=1}^{C} \exp[w_c^\top (A\psi(x_i))]} \right]$$

$$= -\frac{1}{N} \sum_{i=1}^{N} [w_{y_i}^\top A\psi(x_i) - \log \sum_{c=1}^{C} \exp(w_c^\top A\psi(x_i))]. \tag{19}$$

Let $a_j$ represent the $j$th row of $A$. Taking the gradient of $\mathcal{L}$ wrt $a_j$:

$$\nabla_{a_j} \mathcal{L} = -\frac{1}{N} \sum_{i=1}^{N} \left[ w_{y_i}(j)\psi(x_i) - \frac{\sum_{c=1}^{C} \exp[w_c^\top (A\psi(x_i))]w_c(j)\psi(x_i)}{\sum_{c'=0}^{C} \exp(w_{c'}^\top (A\psi(x_i)))} \right]$$

$$= -\frac{1}{N} \sum_{i=1}^{N} \left[ w_{y_i}(j) - \frac{\sum_{c=1}^{C} \exp[w_c^\top f_i]w_c(j)}{\sum_{c'=0}^{C} \exp(w_{c'}^\top f_i)} \right] \psi(x_i)$$

$$= -\frac{1}{N} \sum_{i=1}^{N} \left[ w_{y_i}(j) - \mathbb{E}(w(j)|f_i) \right] \psi(x_i). \tag{20}$$

The gradient update step for $a_j$ is

$$a_{j,\ell+1} = a_{j,\ell} - \alpha \nabla_{a_j} \mathcal{L}$$

$$= a_{j,\ell} + \frac{\alpha}{N} \sum_{i=1}^{N} \left[ w_{y_i}(j) - \mathbb{E}(w(j)|f_i) \right] \psi(x_i).$$

Using the GD update rules for $\{a_j\}_{j=1,d'}$, we have

$$f_{j,\ell+1} = \begin{pmatrix} a_{1,\ell+1}^\top \psi(x_j) \\ \vdots \\ a_{d',\ell+1}^\top \psi(x_j) \end{pmatrix}$$

$$= f_{j,\ell} + \frac{\alpha}{N} \sum_{i=1}^{N} [w_{y_i} - \mathbb{E}(w|f_{i,\ell})] \kappa(x_i, x_j). \tag{21}$$

In (21) it was assumed that the learning rate $\alpha$ was the same for all components of $f$. In general, we may have a different learning rate for each component. Defining $\Lambda = \text{diag}(\alpha_1/N, \ldots, \alpha_{d'}/N)$, we may generalize (21) as

$$f_{j,\ell+1} = f_{j,\ell} + \Lambda \sum_{i=1}^{N} [w_{y_i} - \mathbb{E}(w|f_{i,\ell})] \kappa(x_i, x_j). \tag{22}$$

More generally, $\Lambda$ need not be diagonal. Consider replacing the Euclidean inner product on $\mathbb{R}^{d'}$ with a Mahalanobis inner product $\langle u, v \rangle_M = u^\top M^{-1} v$, where $M \succ 0$. Steepest descent with respect to this geometry yields the *preconditioned* functional gradient descent update

$$f_{j,\ell+1} = f_{j,\ell} + \Lambda \sum_{i=1}^{N} [w_{y_i} - \mathbb{E}(w|f_{i,\ell})] \kappa(x_i, x_j). \tag{23}$$

where $\Lambda \in \mathbb{R}^{d' \times d'}$ is now a full (non-diagonal) matrix. This is a *variable metric* method (Bertsekas, 1999): the RKHS with scalar kernel $\kappa$ remains unchanged, but the descent direction is preconditioned by $\Lambda$, which rotates and scales the functional residual $w_{y_i} - \mathbb{E}(w|f_{i,\ell})$ before it is accumulated into the update. When $\Lambda$ approximates the inverse Hessian of the loss with respect to $f$, this recovers a Newton-type method; when it

approximates the inverse Fisher information matrix, this corresponds to a *natural gradient* method (Amari, 1998).

We note that the matrix-valued kernel perspective provides an alternative justification for (23). Replacing the scalar kernel $\kappa(x_i, x_j) \cdot I_{d'}$ with the separable matrix-valued kernel $K(x_i, x_j) = \kappa(x_i, x_j) \cdot B$, where $B \succ 0$, yields an *intrinsic coregionalization model* (Micchelli & Pontil, 2005; Álvarez et al., 2012) in which cross-component coupling is encoded directly in the RKHS norm. Functional gradient descent in this vector-valued RKHS produces updates of the same form as (23) with $\Lambda \propto B$. However, this changes the function class itself (and hence the implicit regularization), rather than only the optimization geometry. In what follows, we adopt the preconditioning interpretation, as it preserves the original RKHS while allowing $\Lambda$ to be learned.

Finally, there is no reason to constrain $\Lambda$ to be the same across iterations. We therefore allow $\Lambda$ to depend on the iteration index $l$, yielding the update

$$f_{j,\ell+1} = f_{j,\ell} + \Lambda_\ell \sum_{i=1}^{N} \left[ w_{y_i} - \mathbb{E}(w|f_{i,\ell}) \right] \kappa(x_i, x_j). \tag{24}$$

where $\Lambda_\ell \in \mathbb{R}^{d' \times d'}$ is a learned preconditioner specific to iteration $l$. This is naturally motivated by the *algorithm unrolling* (or "learning to optimize") framework (Gregor & LeCun, 2010; Chen et al., 2018): one unrolls the iterative algorithm (23) for $L$ steps, treats the per-step preconditioners $\{\Lambda_\ell\}_{l=1}^{L}$ as learnable parameters, and optimizes them end-to-end with respect to a terminal loss. Since the optimal preconditioner generally depends on the current iterate $f^{(l)}$–which varies systematically across iterations–allowing $\Lambda_\ell$ to differ across layers enables each step to adapt its descent geometry to the stage of the optimization.

If we now generalize this setup, to assume that $f(x)$ may be expressed in terms of $H$ kernels, we have

$$f(x) = \sum_{h=1}^{H} A_h \psi_h(x). \tag{25}$$

We again learn each $A_h$ via gradient descent in the Hilbert space, yielding

$$f_{i,\ell+1} = f_{i,\ell} + \sum_{h=1}^{H} \Lambda_{h,\ell} \sum_{j=1}^{N} [w_{y_j} - \mathbb{E}(w|f_{j,\ell})] \kappa_h(x_i, x_j), \tag{26}$$

where $\kappa_h(x_i, x_j) = \psi_h(x_i)^\top \psi_h(x_j)$. This is the statement of Proposition 1.

## F  Transformer Parameters for Multi-Step GD Via Self-Attention & MLP Layers

The input to the Transformer at layer $l$ is

$$z_{i,\ell} = \begin{pmatrix} f_{i,\ell} \\ \mathbb{E}(w|f_{i,\ell}) \\ w_{y_i} \\ x_i \end{pmatrix}. \tag{27}$$

Within $z_{i,\ell}$, the vector component $f_{i,\ell}$ is iteratively updated with increasing layer index $l$, with the update manifested by each self-attention layer. The expectation $\mathbb{E}(w|f_{i,\ell})$ is (in principle) updated by each MLP layer; the Transformer may *learn* a nonlinear function that is not exactly this expectation. Vector components $f_{i,\ell}$ and $\mathbb{E}(w|f_{i,\ell})$ occupy what we term as *computational scratch space*. The covariates $x_i$ and embedding vector $w_{y_i}$ represent the encoding of the data $(x_i, y_i)$, and the portion of $z_{i,\ell}$ occupied by $(x_i, w_{y_i})$ remains fixed at all Transformer layers.

Each attention block consists of a self-attention layer, composed of two attention heads; one of these attention heads implements $f_{i,\ell} \to f_{i,\ell+1}$ like above (for which $\mathbb{E}(w|f_{i,\ell})$ is needed), and the second attention head erases $\mathbb{E}(w|f_{i,\ell})$, preparing for its update by the subsequent MLP layer.

### F.1 Self-attention layer

In matrix form, the input at layer $l$ is

$$\begin{pmatrix} f_{1,\ell} & \dots & f_{N,\ell} & f_{N+1,\ell} \\ \mathbb{E}(w|f_{1,\ell}) & \dots & \mathbb{E}(w|f_{N,\ell}) & \mathbb{E}(w|f_{N+1,\ell}) \\ w_{y_1} & \dots & w_{y_N} & 0_{d'} \\ x_1 & \dots & x_N & x_{N+1} \end{pmatrix}. \tag{28}$$

The update equation for $f_{i,\ell+1}$ is given by

$$f_{i,\ell+1} = f_{i,\ell} + \Delta f_{i,\ell}, \tag{29}$$

where

$$\Delta f_{i,\ell} = \frac{\alpha}{N} \sum_{j=1}^{N} (w_{y_j} - \mathbb{E}(w|f_{j,\ell}))\kappa(x_i, x_j). \tag{30}$$

#### F.1.1 Self-Attention Head 1

We design $W_K^{(1)}$, $W_Q^{(1)}$, and $W_V^{(1)}$ such that

$$W_K^{(1)} z_{i,\ell} = (0_{d'}, 0_{d'}, 0_{d'}, x_i)^\top \tag{31}$$

$$W_Q^{(1)} z_{j,\ell} = (0_{d'}, 0_{d'}, 0_{d'}, x_j)^\top \tag{32}$$

$$W_V^{(1)} z_{i,\ell} = \left( \frac{\alpha}{N}[w_{y_i} - \mathbb{E}(w|f_{i,\ell})], 0_d, 0_{d'}, 0_{d'} \right)^\top, \tag{33}$$

The output of this first attention head, at position $j \in \{1, \dots, N+1\}$ is

$$\left( \frac{\alpha}{N} \sum_{i=1}^{N} (w_{y_i} - \mathbb{E}(w|f_{i,\ell}))\kappa(x_i, x_j), 0_d, 0_{d'}, 0_{d'} \right)^\top. \tag{34}$$

The output of this first attention head at this first attention layer (before adding the skip connection) is

$$O^{(1)} = \begin{pmatrix} \Delta f_{1,\ell} & \dots & \Delta f_{N,\ell} & \Delta f_{N+1,\ell} \\ 0_d & \dots & 0_d & 0_d \\ 0_{d'} & \dots & 0_{d'} & 0_{d'} \\ 0_{d'} & \dots & 0_{d'} & 0_{d'} \end{pmatrix}. \tag{35}$$

#### F.1.2 Self-Attention Head 2

With the second attention head we want to add $(0_d, -\mathbb{E}(w|f_{j,\ell}), 0_{d'}, 0_{d'})^\top$ from position $j$, so we clear out the prior expectation. This will provide "scratch space" into which, with the next attention layer type, we will update the expectation, using $f_{j,\ell+1}$. To do this, we design $W_Q^{(2)}$ and $W_K^{(2)}$ such that

$$W_K^{(2)} z_{i,\ell} = \lambda(0_{d'}, 0_{d'}, 0_{d'}, x_i)^\top \tag{36}$$

$$W_Q^{(2)} z_{j,\ell} = \lambda(0_{d'}, 0_{d'}, 0_{d'}, x_j)^\top, \tag{37}$$

where $\lambda \gg 1$. With an RBF kernel, for example (similar things will happen with softmax), if $\lambda$ is very large,

$$\kappa(W_K^{(2)} z_{i,\ell}, W_Q^{(2)} z_{j,\ell}) = \delta_{i,j}, \tag{38}$$

where $\delta_{i,j} = 1$ if $i = j$, and it's zero otherwise.

The value matrix is designed as

$$W_V^{(2)} z_{i,\ell} = (0_d, \mathbb{E}(w|f_{i,\ell}), 0_{d'}, 0_{d'})^\top. \tag{39}$$

The output of this head is

$$O^{(2)} = \begin{pmatrix} 0_d & \dots & 0_d & 0_d \\ \mathbb{E}(w|f_{1,\ell}) & \dots & \mathbb{E}(w|f_{N,\ell}) & \mathbb{E}(w|f_{N+1,\ell}) \\ 0_{d'} & \dots & 0_{d'} & 0_{d'} \\ 0_{d'} & \dots & 0_{d'} & 0_{d'} \end{pmatrix}. \tag{40}$$

We then add $P^{(1)}O^{(1)} + P^{(2)}O^{(2)}$, with $P^{(1)}$ and $P^{(2)}$ designed so as to yield the cumulative output of the attention

$$O^{(\text{total})} = \begin{pmatrix} \Delta f_{1,\ell} & \dots & \Delta f_{N,\ell} & \Delta f_{N+1,\ell} \\ -\mathbb{E}(w|f_{1,\ell}) & \dots & -\mathbb{E}(w|f_{N,\ell}) & -\mathbb{E}(w|f_{N+1,\ell}) \\ 0_d & \dots & 0_d & 0_d \\ 0_{d'} & \dots & 0_{d'} & 0_{d'} \end{pmatrix}. \tag{41}$$

This is now added to the skip connection, yielding the total output of this attention layer as

$$T = \begin{pmatrix} f_1^{(\ell+1)} & \dots & f_N^{(\ell+1)} & f_{N+1}^{(\ell+1)} \\ 0_{d'} & \dots & 0_{d'} & 0_{d'} \\ w_{y_1} & \dots & w_{y_N} & 0_{d'} \\ x_1 & \dots & x_N & x_{N+1} \end{pmatrix}. \tag{42}$$

With the first attention layer, with two heads, we update the functions, and we also erase the prior expectations. In the next attention layer, we update the expectations, and place them in the locations of the prior expectations.

## F.2 Multi-Layer Perceptron (MLP) Layer

The vectors connected to $T$ above will go into the next layer, which will be characterized by a MLP. Ideally, the MLP should implement the function (or a related nonlinear function)

$$\mathbb{E}(w|f_{i,\ell+1}) = \sum_{c=1}^{C} w_c \left[ \frac{\exp(w_c^\top f_{i,\ell+1})}{\sum_{c'=1}^{C} \exp(w_{c'}^\top f_{i,\ell+1})} \right], \tag{43}$$

to be consistent with functional GD. Let $g_\gamma(f_{i,\ell+1})$ represent an MLP with parameters $\gamma$. The same MLP acts on each of the vectors at positions $i = 1, \dots, N$, corresponding to the first $N$ columns of $T$, from left. The components of that vector corresponding to $f_{i,\ell+1}$ are input to $g_\gamma(\cdot)$, and the output is a $d'$-dimensional vector. The output is placed in the position of the zeros in $T$.

At each layer of the Transformer, the form of the function in (43) is the same. Consequently, within the Transformer implementation, we tie the MLP parameters across all Transformer layers. In all experiments, the MLP construction is as follows:

- Linear Layer (input dim=5, output dim=10).

- GELU – non linear activation.

- Linear Layer (input dim=10, output dim=5).

In other words, there is one hidden nonlinear layer, with Gaussian Error Linear Unit (GELU) pointwise activations.

# G  Setup for multiple questions with categorical answers

Consider data of the form $(x_i, y_i)$, where $x_i \in \mathbb{R}^d$ are covariates, and $y_i \in \{1, \ldots, C\}^Q$, with $y_i$ representing answers to $Q$ questions with $C$ categorical answers. A special case is $C = 2$, corresponding to $Q$ yes/no questions. Let $y_{i,m} \in \{1, \ldots, C\}$ represent the $q$th component of $y_i$, $q \in \{1, \ldots, Q\}$. We assume that the data are generated from the model

$$p(y_{i,q} = c | X = x_i) = \frac{\exp(f(x_i)^\top w_c^{(q)})}{\sum_{c'=1}^{C} \exp(f(x_i)^\top w_{c'}^{(q)})}, \tag{44}$$

where $\{w_c^{(q)}\}_{q=1,Q}$ represent a set of fixed (learned) vectors, with each $w_c^{(q)} \in \mathbb{R}^{d'}$, and $f(x) \in \mathbb{R}^{d'}$ is a context-dependent latent function. This generalizes our prior setup, which only considered one categorical observation for each $x_i$, to now consider $M$ such categorical observations.

Assume that we are given contextual data $\{(x_i, y_i)\}_{i=1,N}$, from which we wish to infer $f(x)$, and thereby predict $y_{N+1}$ for a query $x_{N+1}$. Assuming that the $Q$ categorical observations are conditionally independent given $f(x)$, as reflected in (44), then the log-likelihood of the contextual data is

$$\mathcal{L} = \sum_{i=1}^{N} \sum_{q=1}^{Q} \log \left[ \frac{\exp(f(x_i)^\top w_c^{(q)})}{\sum_{c'=1}^{C} \exp(f(x_i)^\top w_{c'}^{(q)})} \right] \tag{45}$$

$$= \sum_{i=1}^{N} \sum_{q=1}^{Q} \left[ f(x_i)^\top w_c^{(q)} - \log \sum_{c'=1}^{C} \exp(f(x_i)^\top w_{c'}^{(q)}) \right]. \tag{46}$$

We assume that $f(x) = A\psi(x)$, where $\psi(x) : \mathbb{R}^d \to \mathbb{R}^D$ is a (generally) nonlinear transformation of the covariates $x$ to a $D$-dimensional feature space (which could be infinite dimensional), and $A \in \mathbb{R}^{d' \times D}$ is a latent matrix. The matrix $A$ is context-dependent, while $\psi(x)$ is context-independent. We wish to perform context-dependent gradient ascent to infer $A$.

Gradient ascent applied to this setup yields the following update equation for the latent function:

$$f_{\ell+1}(x) = f_\ell(x) + \alpha \sum_{i=1}^{N} \sum_{q=1}^{Q} \left[ w_{y_{i,q}}^{(q)} - \mathbb{E}(w^{(q)} | f_\ell(x_i)) \right] \kappa(x, x_i), \tag{47}$$

where $\kappa(x_i, x_j) = \psi(x_i)^\top \psi(x_j)$, and

$$\mathbb{E}(w^{(q)} | f_\ell(x_i)) = \frac{\sum_{c=1}^{C} w_c^{(q)} \exp(f_\ell(x_i)^\top w_c^{(q)})}{\sum_{c'=1}^{C} \exp(f_\ell(x_i)^\top w_{c'}^{(q)})}. \tag{48}$$

One can rewrite (47) as

$$f_{\ell+1}(x) = f_\ell(x) + \alpha \sum_{i=1}^{N} \frac{1}{Q} \sum_{q=1}^{Q} \left[ w_{y_{i,q}}^{(q)} - \mathbb{E}(w^{(q)} | f_\ell(x_i)) \right] \kappa(x, x_i) \tag{49}$$

$$= f_\ell(x) + \alpha \sum_{i=1}^{N} \left[ \bar{w}_i - \mathbb{E}(w | f_\ell(x_i)) \right] \kappa(x, x_i), \tag{50}$$

with

$$\bar{w}_i = \frac{1}{Q} \sum_{q=1}^{Q} w_{y_{i,q}}^{(q)}, \qquad \mathbb{E}(w | f_\ell(x_i)) = \frac{1}{Q} \sum_{q=1}^{Q} \mathbb{E}(w^{(q)} | f_\ell(x_i)), \tag{51}$$

where we see that $\mathbb{E}(w | f_\ell(x_i))$ is an average over expectations.

## H  Linearization of the Expectation

Consider

$$
\mathbb{E}(w|f_{i,\ell}) = \mathbb{E}(w|f_{i,\ell-1} + \Delta f_{i,\ell-1}) \tag{52}
$$

$$
= \frac{\sum_{c=1}^{C} \exp[w_c^\top f_{i,\ell-1} + w_c^\top \Delta f_{i,\ell-1}]w_c}{\sum_{c'=1}^{C} \exp(w_{c'}^\top f_{i,\ell-1} + w_{c'}^\top \Delta f_{i,\ell-1})}. \tag{53}
$$

Assuming that $\Delta f_{i,\ell-1}$ makes a small change relative to $f_{i,\ell-1}$, which can be controlled by the learning rate, we may approximate $\mathbb{E}(w|f_{i,\ell})$ by its first-order (linear) Taylor expansion. We have

$$
\nabla_f \frac{\exp(w_c^\top f)}{\sum_{c'=1}^{C} \exp(w_{c'}^\top f)} = w_c \frac{\exp(w_c^\top f)}{\sum_{c'=1}^{C} \exp(w_{c'}^\top f)} - \frac{\exp(w_c^\top f)}{[\sum_{c'=1}^{C} \exp(w_{c'}^\top f)]^2} \sum_{c'=1}^{C} w_{c'} \exp(w_{c'}^\top f)
$$

$$
= \frac{\exp(w_c^\top f)}{\sum_{c'=1}^{C} \exp(w_{c'}^\top f)} \Big[ w_c - \mathbb{E}(w|f) \Big]. \tag{54}
$$

Therefore

$$
\mathbb{E}(w|f_{i,\ell}) \approx \mathbb{E}(w|f_{i,\ell-1}) + \sum_{c=1}^{C} w_c \frac{\exp(w_c^\top f_{i,\ell-1})}{\sum_{c'=1}^{C} \exp(w_{c'}^\top f_{i,\ell-1})}[w_c - \mathbb{E}(w|f_{i,\ell-1})]^\top \Delta f_{i,\ell-1} \tag{55}
$$

$$
= \mathbb{E}(w|f_{i,\ell-1}) + W_e \cdot \mathrm{diag}[\mathrm{softmax}(W_e^\top f_{i,\ell-1})] \cdot \tilde{W}_{e,\ell-1}^\top \cdot \Delta f_{i,\ell-1}, \tag{56}
$$

where the $c$th column of $\tilde{W}_{e,\ell-1}$ corresponds to $w_c - \mathbb{E}(w|f_{i,\ell-1})$.

For $\ell = 1$, $f_{i,\ell-1} = 0_{d'}$, and therefore $\mathrm{softmax}(W_e^\top f_{i,\ell-1})$ is a uniform $C$-dimensional probability mass function (PMF), and hence

$$
\mathbb{E}(w|f_{i,1}) \approx \frac{1}{C} \Big[ \sum_{c=1}^{C} w_c + W_e \tilde{W}_{e,0}^\top \Delta f_{i,0} \Big], \tag{57}
$$

where the $c$th column of $\tilde{W}_{e,0}$ is $w_c - \frac{1}{C} 1_C$, where $1_C$ is a $C$-dimensional vector of all ones.

For $\ell = 1$, for each $i$ we may approximate $\mathbb{E}(w|f_{i,1}) \approx \mathbb{E}(w|f_{i,0}) + M_1 \Delta f_{i,0}$ where $M_1 = \frac{1}{C} W_e \tilde{W}_e^\top$. Note that this matrix is independent of $i$. More generally, and with a weaker approximation, we use $\mathbb{E}(w|f_{i,\ell}) \approx \mathbb{E}(w|f_{i,\ell-1}) + M_\ell \Delta f_{i,\ell-1}$. For $\ell > 1$ this approximation is less appropriate, because the above first-order analysis indicates that the linear approximation is *dependent on $i$*, which we are ignoring. We expect the linear approximation to work best when the first two steps of functional GD reach near convergence of GD-based inference.

## I  GD Parameters for Attention-Only Transformer

Under the linear approximation for $\mathbb{E}(w|f_{i,\ell})$, we can implement Transformer-based inference with a single attention head, as detailed below. Let $\bar{w} = \frac{1}{C} \sum_{c=1}^{C} w_c$, i.e., the average embedding vector for all $C$ categories. At the input layer, consider the encoding $(x_i, w_{y_i} - \bar{w}, 0_{d'})$, for $i = 1, \ldots, N$, where the $0_{d'}$ vector is positioned where $f_i$ will be updated. Recall that $\bar{w} = \mathbb{E}(w|0_{d'})$, which is the zero-order (initial) approximation to the expectation. The query is encoded as $(x_{N+1}, 0_{d'}, 0_{d'})$.

The Transformer matrices at each layer are

$$
W_Q = W_K = \begin{pmatrix} I_{d \times d} & 0_{d \times d'} & 0_{d \times d'} \\ 0_{d' \times d} & 0_{d' \times d'} & 0_{d' \times d'} \\ 0_{d' \times d} & 0_{d' \times d'} & 0_{d' \times d'} \end{pmatrix}, \tag{58}
$$

such that $W_Q z_{i,\ell} = W_K z_{i,\ell} = (x_i, 0_{d'}, 0_{d'})$ at each layer, where $z_{i,\ell} \in \mathbb{R}^{d+2d'}$ is the vector at position $i$, output from layer $\ell$.

The $W_V$ matrix can be expressed as

$$W_V = \frac{\alpha}{N} \begin{pmatrix} 0_{d\times d} & 0_{d\times d'} & 0_{d\times d'} \\ 0_{d'\times d} & 0_{d'\times d'} & 0_{d'\times d'} \\ 0_{d'\times d} & I_{d'\times d'} & 0_{d'\times d',} \end{pmatrix} \tag{59}$$

and position $i = 1, \ldots, N$ are used for keys and values, where all $i = 1, \ldots, N + 1$ positions are used as queries. The output of attention from layer-1 is

$$\sum_{i=1}^{N} W_V z_{i,0} \kappa(W_K z_{i,0}, W_Q z_{j,0}) = \begin{pmatrix} 0_d \\ 0_{d'} \\ \frac{\alpha}{N} \sum_{i=1}^{N} [w_{y_i} - \bar{w}] \kappa(x_i, x_j) \end{pmatrix}, \tag{60}$$

where it is understood that the above $d'$-dimensional vector $v_j$ is positioned as $(0_d, 0_{d'}, v_j)$ in the coordinate system of the Transformer vectors.

Finally, there is an output projection matrix at each layer:

$$P_\ell = \begin{pmatrix} 0_{d\times d} & 0_{d\times d'} & 0_{d\times d'} \\ 0_{d'\times d} & 0_{d'\times d'} & -M_\ell \\ 0_{d'\times d} & 0_{d'\times d'} & I_{d'\times d'} \end{pmatrix}, \tag{61}$$

which will update the expectation, to within a linear approximation, with the updated increment to the expectation appended to $\bar{w}$:

$$z_{j,0} + P_1 \sum_{i=1}^{N} W_V z_{i,0} \kappa(W_K z_{i,0}, W_Q z_{j,0}) = \begin{pmatrix} x_j \\ w_{y_j} - \bar{w} - M_1 \Delta f_{j,0} \\ \Delta f_{j,0} \end{pmatrix}, \tag{62}$$

where $\Delta f_{j,0} = \frac{\alpha}{N} \sum_{i=1}^{N} [w_{y_i} - \bar{w}] \kappa(x_i, x_j)$.

The latent function $f_{i,\ell}$, for $i = 1, \ldots, N + 1$, is updated in the last $d'$ positions of $z_{i,\ell}$ at each layer $\ell$, and this is sent into the softmax at the last layer (for position $i = N + 1$).

## J  Distribution Mismatch: Performance Comparison across different variants of GD models

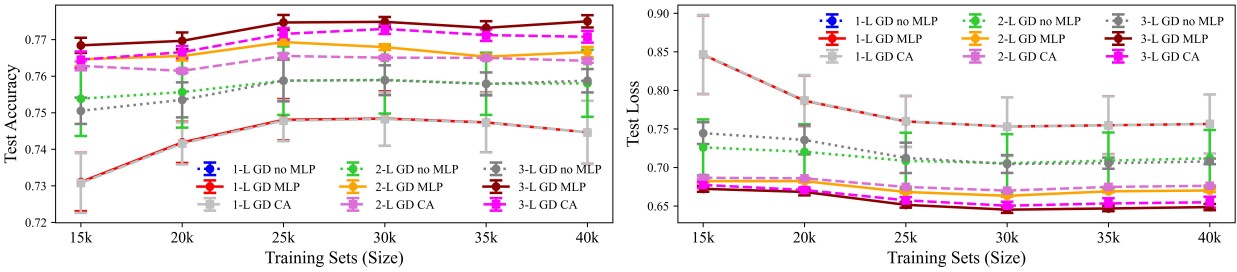

Figure 8: (Partial zoomed-in plot) ICL classification accuracy (left) and loss (right) on TinyImageNet (test-data), with the Transformer models trained on ImageNet1k (training-data), as a function of the number of training contexts $M$. All GD models use sparse parameterization guided by the analysis in Section 2. Results are shown for 1, 2, and 3 layer(s) GD models.

