# OpenReview forum: "On the Role of MLP Layers in Transformer ICL with Categorical Outcomes"
_TMLR — Decision pending for TMLR_

### Review · Reviewer_3dXo · 2026-03-15

**Summary Of Contributions:**

1) Study of ICL in Transformers on categorical outputs -> MLPs play a natural role in doing a necessary non-linear computation to implement GD.
2) Although MLPs can compute this non-linear part, the authors show that softmax self-attention could and does, in practice, perform this computation.
3) Although the picture is quite complicated, the authors try to dissect the role of MLPs in transformers when trained on synthetic as well as some real-worldish ICL tasks.

**Additional Comments:**

Generally, I think this paper is very borderline because it is tackling a quite narrow problem and the authors show (but are honest about it) quite mixed results. These could and should be improved quite a bit nevertheless.

**Audience:**

Yes

**Audience Explanation:**

I think there is some audience for this research, although the studied problem is quite narrow by design.

**Claims And Evidence:**

Yes

**Claims Explanation:**

The paper does quite a thorough and honest investigation of their claims. Experiments are quite detailed, although this should be extended quite significantly. See below.

**Requested Changes:**

MLPs are universal function approximators and can compute "anything"; it is unclear what computation is performed in more realistic settings. I think this is still unclear and only little evidence for their actual role in the more real-world experiments. In https://arxiv.org/abs/2205.05055 (not cited), the authors contrast between in-weights and in-context learning. Unfortunately, this topic is not discussed, and only the search for a specific quantity (E(w|f) is investigated. This is investigation is quite narrow and also not very conclusive.

In previous works that showed that a single layer of softmax can perform a single step of GD, the authors started by showing that, actually, next token prediction of a single layer with one single head does perform one step of GD. Is this the case here? I think this is not shown in the paper? So what happens when you compare one layer/head of softmax (+MLP layer) vs one step of GD.
Sorry if I missed this.
Based on this, the authors then proceeded to the multi-layer case.

Other investigations: Which role does the width of the MLP have?

---

> ### Author Response · Authors · 2026-03-23
> **Response to Reviewer 3dXo**
>
> We thank the reviewer for the careful reading and for bringing an important related paper to our attention. We address each point below.
>
> # On the narrowness of the investigation and the role of $E(w|f)$
>
> We wish to respectfully push back on the suggestion that our investigation is narrow or inconclusive. We believe the referenced paper (Chan et al., 2022, https://arxiv.org/abs/2205.05055) actually helps illustrate the breadth of our contribution, and we thank the reviewer for drawing our attention to it — we will add it to our references and discuss it in the revision.
>
> That paper demonstrates empirically that burstiness — the presence of multiple context examples sharing the label of the query — is critical for ICL to emerge in Transformers applied to categorical outcomes. However, it does not explain why burstiness matters. Our theory provides exactly this explanation. In our framework, the context examples with the same label as the query will have covariates $x_j$ similar to the query covariate $x_i$, and therefore $\kappa (x_i, x_j)$ will be large, producing strong attention to those examples. The term $w_{y_j}$ in Equation (5) is the embedding vector for label $y_j$; under burstiness, the weighted accumulation of error signals strongly aligns the predictive function with the embedding vector of the query label, yielding accurate prediction. Our theory thus explains mechanistically why burstiness is essential — a question left open by the referenced paper.
>
> We also note the structural parallel between our setup and theirs: in both cases images are encoded via a deep model and categorical tokens are mapped to learned embedding vectors. Our work provides the theoretical foundation that complements their empirical observations. We will add a detailed discussion of this connection in the revision.
>
> Furthermore, ours is the first theoretical ICL paper to examine the role of MLP layers — prior work exclusively considered real-valued outputs, for which no MLP role exists. The referenced paper considers categorical outputs but provides no theoretical account of the Transformer's mechanism. We believe these two points together argue against the characterization of our contribution as narrow.
>
> # On whether a single attention layer performs one step of GD
>
> The answer is yes, and we agree we should make this more prominent in the revision. As stated on page 7, in the paragraph beginning with the bolded sentence "The first two steps...": with initialization $f_{i,0} = 0$, the first GD step can be performed exactly with the first self-attention layer. Figure 1 confirms empirically that the one-layer GD model and Trained TF align well, and Figure 5 shows that the learned parameters of Trained TF match the theoretical GD construction.
>
> Regarding the natural follow-up question — what happens with one attention layer followed by one MLP layer — the answer follows directly from our framework. The MLP's role is to update the expectation $E(w|f_{i,\ell})$ in preparation for the next attention layer. After a single attention layer, if there is no subsequent attention layer, the updated expectation has no downstream effect on the prediction. Therefore, adding an MLP after a single attention layer has no impact on the output in our formulation. This is consistent with our theoretical framework and we will make this point explicit in the revision.
>
> # On the width of the MLP
>
> This is examined directly in Figure 4 of the paper (Section 5.3), which shows how accurately the MLP approximates the expectation $E(w|f_{i,\ell})$ as a function of the number of hidden units. The key finding is that the approximation error diminishes quickly with width, and that a hidden dimension equal to or larger than the rank of the token embedding matrix is sufficient. The phenomenon is consistent across all experiments we have conducted. We will add further results on MLP width to the Appendix if the reviewer would find this useful.
>
> # On in-weights versus in-context learning
>
> We thank the reviewer for raising this distinction, as discussed in the referenced paper. Our paper focuses exclusively on ICL — we do not consider the in-weights regime. We will add a discussion in the revision connecting our theoretical framework to the in-weights versus in-context distinction, and clarifying that our contribution is specifically to the ICL side of this dichotomy.

---

### Review · Reviewer_qUKY · 2026-04-12

**Summary Of Contributions:**

The paper addresses a gap in mechanistic understanding of Transformers: prior work showed attention alone suffices for ICL with real-valued outputs, but most practical tasks involve categorical outputs. The authors show that categorical outputs introduce a nonlinear interlayer computation recomputing the softmax-weighted expected embedding $E(w|f)$ at each layer that attention cannot perform but MLP layers naturally approximate.

1. **Concrete role for MLP layers**: Categorical ICL requires recomputing a nonlinear expectation under the softmax between layers. MLP layers are the natural Transformer component to approximate this. This role is absent for real-valued outputs.

2. **Conditions for attention-only sufficiency**: A linear approximation to $E(w|f)$ is valid at initialization (softmax in linear regime, all positions share similar representations). Attention-only works for ~2 layers under matched distributions; breaks down at greater depth or under distribution mismatch.

3. **Sparse GD parameterization**: Theory-guided parameter reduction (~50×: ~300 vs ~15,000 parameters for a 3-layer model), achieving the same accuracy as an unconstrained Transformer with ~5× fewer training examples.

4. **Applications**: Multi-question categorical ICL applied to surgical action triplet recognition (CholecT45), achieving mAP > 0.9 with sparse models in a data-scarce domain.

**Audience:**

Yes

**Audience Explanation:**

This paper fits TMLR's scope squarely:
1. Answers a concrete open architectural question (what do MLPs do?) in the active mechanistic interpretability literature.

2. Extends the functional gradient descent framework to categorical outputs, which is a natural and valuable step.

3. Sparse parameterization has immediate practical value in data-limited settings (medical imaging, scientific domains).

4. Real-world surgical application distinguishes this from purely synthetic mechanistic ICL work.

**Broader Impact Concerns:**

1. Abstract framing may lead practitioners to over-generalize findings to large LLMs where evidence is absent.

2. Hand-derived sparse parameterization may fail silently when data deviates from assumed ICL structure; failure modes not discussed.

3. Surgical application is proof-of-concept only; clinical use would require extensive additional validation.

**Claims And Evidence:**

Yes

**Claims Explanation:**

1. **Synthetic experiments (Section 5.1)**: Data efficiency gap between GD (sparse) and Trained TF is striking and reproducible across 5 seeds and 1–3 layer models.

2. **Distribution mismatch (Section 5.2)**: Training on Caltech256, testing on TinyImageNet and five DomainNet domains. Degradation of attention-only models specifically at 2+ layers precisely matches theoretical predictions.

3. **MLP replacement experiment (Section 5.3)**: Replacing cross-attention with an MLP and retraining only those parameters directly supports the claim that MLPs approximate $E(w|f)$.

4. **Parameter visualization (Figure 5)**: Unconstrained Trained TF converges to the same $Q^TK$ and $PV$ structure as the theory-derived GD model.

5. **Surgical application (Section 5.5)**: mAP > 0.9 with 2-layer models; comparison between MLP and attention-only validates the matched-distribution prediction.

**Requested Changes:**

### Major
1. **Missing formal result on MLP convergence**: No proof that trained MLPs converge to approximating $E(w|f)$. Even a partial formal result or training-time measurements of MLP output vs. $E(w|f)$ would strengthen the paper.
2. **Scale gap**: All experiments use 1–3 layer models. Experiments at 6–12 layers, even synthetic, are needed to support claims relevant to modern Transformers.
3. **Proposition 1 assumes RKHS structure**: This additional assumption is not required for the main argument. Clarify whether main results depend on it.
4. **Missing Trained TF baseline in distribution mismatch**: Figures 2–3 omit the unconstrained Trained TF. Including it would directly test whether data efficiency advantages persist under distribution shift.

### Minor
5. **Quantify the "linear regime" condition**: The criterion for softmax linearity is stated qualitatively; even a rough bound on ‖f‖ would make this actionable.
6. **Figure 4 conflates MLP width and matrix rank**: Needs clarification (e.g., does rank $= d′ = 5$?).
7. **Multi-question extension underanalyzed**: Eq. (15) lacks theoretical development; brief analysis or explicit acknowledgment of future work needed.
8. **Code not yet available**: TMLR emphasizes reproducibility; a blinded repository at submission time is expected.

---

> ### Author Response · Authors · 2026-04-21
> **Response to Reviewer qUKY**
>
> Thank you for the thorough review.
>
> # Major Point 1:
>
> We acknowledge this point honestly: a formal proof that trained MLPs converge to approximating $E(w|f_{i,\ell})$ is a limitation that is not unique to our paper. We flag this explicitly in Section 6.
> Exact empirical demonstration faces a fundamental difficulty: with all other model parameters free to vary, the MLP output and $E(w|f_{i,\ell})$ can differ by transformations that leave overall predictions invariant. This is why, for example, we focus on appropriate matrix *products* in Fig 5 rather than individual parameters.
>
> That said, we have made serious empirical efforts to gather evidence, and we believe the cumulative case is strong:
> •	Sec 5.3 shows directly that an MLP can accurately approximate the cross-attention output that computes $E(w|f_{i,\ell})$, recovering it with very few hidden units. This demonstrates that the MLP has sufficient expressive power for this computation.
> •	Fig 5 and the additional parameter visualizations we will add to the Appendix show that Trained TF consistently learns parameter structure matching the GD construction across multiple settings, with and without MLP layers.
> •	The performance gap between MLP and attention-only models appears precisely where the theory predicts — at greater depth and under distribution mismatch — and is absent where the theory predicts attention-only sufficiency. This is strong indirect evidence that the MLP is performing the functionally predicted role.
>
> We will sharpen the discussion of this limitation in Sec 6 to make the above reasoning explicit.
>
> # Major Point 2
>
> Within the functional GD framework, each Transformer layer corresponds to one refinement step. After approximately 3 steps, functional GD has converged — additional layers produce no further change in the output. This is why the entire ICL-as-GD literature, including von Oswald et al. (2023), Ahn et al. (2023), Cheng et al. (2024), and Wang et al. (2025), considers models of 1-3 layers. Running a 12-layer GD model would simply replicate the 3-layer output at every subsequent layer. The depth range in our experiments is not an oversight but a reflection of the framework's natural scope.
>
> The reviewer's implicit question — what happens in large language models with many layers — is genuinely important and goes beyond the current framework in a specific way. In language models there are no covariates in the ICL sense used here, so pure GD inference cannot account for all layers. We hypothesize that large language models perform two intermingled functions across layers: context-dependent feature extraction from token embeddings, and GD-like inference on those features.
>
> So motivated, we have (separate, follow-on work) constructed a 24-layer *language* model that explicitly separates these two functions — 12 layers dedicated to feature extraction and 12 to GD-like inference — and find that it achieves performance nearly identical to an unconstrained black-box Transformer across multiple language modeling tasks (for models up to 8B parameters). This is an extensive study in its own right and will be reported separately.  If the reviewer feels it would be valuable, we are happy to add a brief forward-looking discussion of this connection to the paper; extension to language supports the ideas in this paper.
>
> # Major Point 3
>
> The reviewer is correct that the RKHS assumption is not required for the main argument. Secs 2.1-2.3 develop the core intuition — prediction error, weighted averaging via attention, nonlinear expectation requiring MLP — entirely without RKHS structure. The RKHS assumption is invoked only in Proposition 1 to provide a formal grounding for the functional GD connection. We further generalize away from this when we use softmax attention, which is not formally a kernel but which naturally implements the weighted averaging step. We will add a clarifying remark at the start of Section 2 and in the Proposition 1 statement to make this dependency explicit.
>
> # Major Point 4
>
> This is a legitimate missing baseline and we thank the reviewer for pointing it out. We will add the unconstrained Trained TF results to Figures 2 and 3 in the revision.
>
> # Minor Points
>
> * We will add a norm bound on $\|f_{i,\ell}\|$ in the appendix that makes the softmax linearity condition more precise and actionable. We will build upon the current Appendix H, which details foundational elements of the linear approximation.
>
> * We will clarify the relationship between MLP hidden units and matrix rank explicitly in the caption and surrounding text.
>
> * We acknowledge that Equation (15) is presented without full theoretical development. We will add an explicit acknowledgment that formal analysis of the multi-question extension is left as future work.
>
> * Code is available at https://anonymous.4open.science/r/code-tf-gd-mlp-4252/ . The link will be updated to the full repository upon publication.
>
> * We will address your comments on broader impact

---

### Review · Reviewer_XeEn · 2026-04-16

**Summary Of Contributions:**

This paper contributes to a line of research dedicated to the mechanistic and theoretical study of supervised transformer meta-learning, motivated by the in-context learning (ICL) phenomenon observed in language models. The main contributions of the current paper are: (1) an argument for a key role played by MLP layers when the target variable is categorical; (2) a first-order Taylor expansion argument for when attention-only models can still approximately solve the task well; (3) a set of explicit predictions related to model depth for when attention-only models break; (4) a sparse transformer parametrization that greatly reduces the number of trainable parameters; (5) experiments on synthetic data, image classification under dataset (meta-training -> meta-testing) mismatch, and on a multi-question answering task.

**Audience:**

Yes

**Audience Explanation:**

The literature devoted to the study of in-context learning toy models is already large, but it continues to be an active topic of investigation of relevance to the TMLR readership.
The findings reported in the paper are incremental, but if more carefully confirmed (or even better yet, expanded), they could be interesting in nuanced ways. The case of categorical targets is highly relevant, and still somewhat understudied, compared to real-valued regression, and the paper brings a little bit more light to it. The distribution shift analyses, using non-synthetic data, are quite interesting and unusual in ICL toy model studies.

**Broader Impact Concerns:**

None.

**Claims And Evidence:**

Yes

**Claims Explanation:**

Positive points:
- The theoretical derivations appear to be correct to me.
- The data-efficiency advantage of the sparse parameterization is supported on synthetic data (results in Fig. 1)

Points for improvement:
- While the DomainNet/TinyImageNet experiments show the predicted degradation-with-depth pattern, there could be other reasons for performance degradation under distribution mismatch. Just as an example, one could imagine that the meta-learned preconditioning doesn’t transfer well. The paper needs additional controls to back its claims up.
- The visualization in Fig. 5 is not nearly enough evidence that the construction holds generally, not even in small models. In particular, the analysis is for attention-only models, but MLPs are crucial for some of the main points of the paper.

**Requested Changes:**

Minimal changes requested:
- Position the paper more precisely against the literature. In particular, compare better to Wang et al. (2025), as it is still not very clear what exactly is new compared to this highly related work; acknowledge that meta-learned preconditioning is common to many previous analyses of ICL in transformers; position findings against prior work studying the role of MLPs (e.g., "In-Context Learning of a Linear Transformer Block: Benefits of the MLP Component and One-Step GD Initialization" by Zhang et al, NeurIPS 2024).
- Discuss broader meta-learning literature outside the ICL toy model niche. For example, Kirsch et al. ("General-Purpose In-Context Learning by Meta-Learning Transformers", NeurIPS Workshops 2022).
- Rule out other sources of performance degradation under distribution mismatch by adding additional controls.
- Gather more empirical evidence for the theoretical claims and constructions in the paper.

---

> ### Author Response · Authors · 2026-04-21
> **Response to Reviewer XeEn**
>
> Thank you for the careful reading and constructive feedback.
>
> # Sources of performance degradation
>
> The reviewer suggests that performance degradation under distribution mismatch could reflect poor transfer of meta-learned preconditioning rather than breakdown of the softmax linear approximation.
>
> The preconditioning interpretation — as formalized in Ahn et al. (2023) — is specific to **linear self-attention**, where the bilinear form of attention weights allows Q and K matrices to encode the inverse covariance of the input distribution. Our models use softmax (or other nonlinear) attention, where Q and K extract covariates to compute a kernel similarity rather than a distributional statistic. The formal mechanism through which preconditioning arises in Ahn et al. does not operate in our setting. We will add a clarifying remark to this effect in the revision.
>
> In our setting, the MLP-equipped and attention-only models share identical **nonlinear** attention mechanisms, the latter particularly distinct from Ahn et al. (2023). The only architectural difference between the two is the presence or absence of MLP layers. Therefore, if preconditioning transfer failure were a significant factor, it would degrade both models equally and could not explain any gap between them. Yet Figs. 2 and 3 show a pronounced gap specifically between the MLP and attention-only models under distribution mismatch, with no such gap under matched distributions (Fig. 1). The comparison between MLP and attention-only models is thus already a clean within-experiment control for preconditioning effects. We agree the argument should be made more explicitly in the paper, and we will clarify this in the revision.
>
> We agree, nevertheless, that it is important to have sharp experiments to manifest mechanistic conclusions with a model as  sophisticated as the Transformer. In the final version, we will add further experiments, as suggested, placed in the Appendix, for space.
>
> # On Fig 5
>
> The reviewer correctly notes that Fig 5 is a single visualization and may not be representative. We have in fact conducted many experiments of this type across varying depths, datasets, and parameterizations — both with and without MLP layers — and the qualitative conclusion is consistent throughout: the Trained TF learns parameter structure closely matching the GD construction, with Q^T K operating on the covariate subspace and PV updating the latent representation as predicted.
>
> We omitted the MLP cases from the main paper because the two-head structure makes the parameter comparison more complicated, as noted in the text. However, we will add these additional results to the Appendix in the revision, along with additional attention-only cases across depths and datasets.
>
> We also point the reviewer to Sec 5.3 as complementary evidence. There, we freeze all parameters of a cross-attention model that computes $E(w|f_{i,\ell})$ exactly, remove the cross-attention, and replace it with either an MLP or a linear approximation trained from scratch. Both recover the cross-attention output accurately with very few units, providing direct evidence that the MLP is well-suited to approximate the specific nonlinear computation our analysis identifies. Together with the Fig 5 parameter structure analysis, this constitutes stronger evidence for the theoretical construction than either experiment alone.
>
> # Positioning against the literature
>
> Wang et al. (2025) identify the need for a nonlinear computation between attention layers but address it by introducing a novel cross-attention mechanism that computes $E(w|f_{i,\ell})$ exactly. Their model does not correspond to the standard Transformer. We analyze how the standard Transformer — with its existing MLP layers — approximates this computation, explain when and why the approximation succeeds or fails, and leverage this understanding for a sparse parameterization with substantial data efficiency. Wang et al. do not consider the standard Transformer with MLPs, do not analyze the conditions under which attention-only models suffice, and do not develop the distribution mismatch predictions.
>
> Zhang et al. (2024) show that MLP layers benefit ICL in a Linear Transformer Block for real-valued linear regression, where the MLP's role is bias correction for non-zero task mean — a shared-signal phenomenon specific to the linear regression setting with linear attention. Our paper identifies a distinct and complementary MLP role that arises from categorical outputs and the softmax nonlinearity, and is entirely absent in the real-valued setting. We will discuss this paper in the revision.
>
> We will acknowledge in the revision that learned preconditioning is a common feature of prior ICL analyses, while noting the distinctions discussed above regarding linear versus softmax attention.
>
> Kirsch et al. (2022). We will add a paragraph in related work situating our contribution within the broader meta-learning perspective on ICL, as suggested.

---

### Decision · Action_Editor_9MTx · 2026-06-11

**Recommendation:** Accept as is

**Audience:**

Yes

**Audience Explanation:**

There is quite some literature around ICL theory and the role of different architecture parts in transformers, to which this paper adds.

**Claims And Evidence:**

Yes

**Claims Explanation:**

All three reviewers agree that the paper's claims are supported by evidence. The paper makes an interesting contribution to the ICL community.